# Transductive and Inductive Outlier Detection with Robust Autoencoders

Ofir Lindenbaum[1*]                    Yariv Aizenbud[2*]                    Yuval Kluger[3,4,5]

[1]Faculty of Engineering, Bar-Ilan University, Ramat Gan, Israel
[2]School of Mathematical Sciences, Tel Aviv University, Tel Aviv, Israel
[3]Program in Applied Mathematics, Yale University, New Haven, CT, USA
[4]Department of Pathology, Yale University School of Medicine, New Haven, CT, USA
[5]Computational Biology & Bioinformatics Program, Yale University, New Haven, CT, USA
[*]These authors contributed equally

## Abstract

Accurate detection of outliers is crucial for the success of numerous data analysis tasks. In this context, we propose the Probabilistic Robust AutoEncoder (PRAE) that can simultaneously remove outliers during training (transductive) and learn a mapping that can be used to detect outliers in new data (inductive). We first present the Robust AutoEncoder (RAE) objective that excludes outliers while including a subset of samples (inliers) that can be effectively reconstructed using an AutoEncoder (AE). RAE minimizes the autoencoder's reconstruction error while incorporating as many samples as possible. This could be formulated via regularization by subtracting an $\ell_0$ norm, counting the number of selected samples from the reconstruction term. As this leads to an intractable combinatorial problem, we propose two probabilistic relaxations of RAE, which are differentiable and alleviate the need for a combinatorial search. We prove that the solution to the PRAE problem is equivalent to the solution of RAE. We then use synthetic data to demonstrate that PRAE can accurately remove outliers in various contamination levels. Finally, we show that using PRAE for outlier detection leads to state-of-the-art results for inductive and transductive outlier detection.

## 1   INTRODUCTION

Unsupervised outlier detection is a critical problem in data mining and machine learning. It involves finding unusual measurements in datasets that have no labeling. Detecting anomalous samples is essential for empirical science in various fields, including biology [Lenning et al., 2018], geophysics [Bregman et al., 2021], engineering, and cybersecurity [Chawla et al., 2019]. Outliers, also known as anomalies, are samples that deviate significantly from the majority of observations. However, defining what is normal and abnormal remains a challenging task in machine learning. Our focus is on unsupervised outlier detection to identify anomalies in two settings: (1) inductive learning, where we want to identify outliers from newly arrived samples without additional training, and (2) transductive learning, aiming to identify and remove outliers from an existing dataset.

One effective way to detect anomalies is by analyzing the density of data. This can be done by estimating the data density followed by detection of anomalies as samples that reside in the low probability density regions [Bishop, 1994]. Density-based models include HBOS [Goldstein and Dengel, 2012], Local Outlier Factor (LOF) [Breunig et al., 2000], or some of its variants [Jin et al., 2001, Tang et al., 2002b, Jin et al., 2006]. More recent probabilistic approaches include [Kriegel et al., 2009b, Constantinou, 2018, Rozner et al., 2024]. Other schemes [Aizenbud et al., 2015, Ramaswamy et al., 2000, Angiulli and Pizzuti, 2002, Ghoting et al., 2008] rely on distances between samples to identify anomalies. The fundamental assumption of these approaches is that normal points have dense neighborhoods, whereas outliers are far from their neighbors. An alternative paradigm for anomaly detection is one-class classification [Chen et al., 2001, Ruff et al., 2018, Perera et al., 2021, Deng and Li, 2022]. In this method, anomalies are identified as samples that significantly deviate from the bulk part of the data, also known as the "one-class".

High-dimensional measurements can often be represented by a low-dimensional subspace or manifold [Aizenbud and Sober, 2021, Roweis and Saul, 2000, Peterfreund et al., 2020]. By assuming that normal samples lie close to a low-dimensional latent manifold, while outliers are diverse and do not conform to the same manifold structure, anomalies can be detected using dimensionality reduction methods like Principal Component Analysis (PCA) [Pearson, 1901] or deep Autoencoders (AE) [Rumelhart et al., 1985, Japkowicz et al., 1995, LeCun et al., 1989]. Robust PCA schemes

[Lerman and Maunu, 2018b] search for a low-dimensional linear subspace that best fits the inliers. These models can simultaneously identify anomalies and learn a reduced subspace, but they are limited to linear transformations. To overcome this limitation, several authors have proposed using AEs [Chen et al., 2017, Zhou and Paffenroth, 2017] to learn a valuable nonlinear mapping for detecting outliers.

Generative models have emerged as powerful tools for learning data distribution, making them useful for identifying anomalies [Zong et al., 2018, Liu et al., 2019, Du and Mordatch, 2019, Eduardo et al., 2020]. In semi-supervised settings, deep neural networks have been utilized by several authors [Hendrycks et al., 2018, Wang et al., 2019, Goyal et al., 2020, Reiss et al., 2021, Hojjati and Armanfard, 2021] to model the normal part of the data and identify outlier samples that differ significantly from normal ones. Recently, self-supervision [Hendrycks et al., 2019, Bergman and Hoshen, 2020] and transfer learning [Deecke et al., 2021] have gained interest in anomaly detection in vision to detect semantic anomalies.

This work focuses on unsupervised anomaly detection for general data (not necessarily images) and proposes a novel Probabilistic Robust autoencoder (PRAE). PRAE can remove outliers during training (transductive) and be used to detect outliers in unseen data (inductive). Our contributions are four folds: (1) We formulate the robust autoencoding (robust-AE) problem by incorporating an $\ell_0$ term penalizing the number of observations included in the AE's reconstruction loss. (2) We propose two probabilistic relaxations for robust AE and demonstrate that they could be effectively trained using standard optimization tools such as stochastic gradient descent (SGD). (3) We show theoretically that the solution of the probabilistic relaxation is equivalent to the solution to the robust-AE problem. (4) We propose two unsupervised schemes to tune the regularization parameter controlling the robustness of the PRAE. (5) We demonstrate, using synthetic and real data with up to $32k$ variables and $1M$ samples, that PRAE outperforms leading anomaly detection methods in multiple settings.

**Notation:** Throughout the paper, we treat anomalies and outliers similarly since we only assume that they deviate from the "normal" samples. Furthermore, we denote vectors using bold lowercase letters such as $\boldsymbol{x}$. Scalars are denoted by lowercase letters such as $y$. The $n^{th}$ vector-valued observation is denoted as $\boldsymbol{x}_n$ while $x[d]$ represents the $d^{th}$ feature (or entry) of the vector $\boldsymbol{x}$. Matrices are denoted by bold uppercase letters $\boldsymbol{X}$. The $\ell_p$ norm of $\boldsymbol{x}$ is denoted by $\|\boldsymbol{x}\|_p$.

# 2 METHOD

## 2.1 ROBUST AUTOENCODER

We observe a set of data samples represented by $\boldsymbol{X} = \{\boldsymbol{x}_1, \ldots, \boldsymbol{x}_N\}$, where each sample $\boldsymbol{x}_i$ is a vector of real numbers in $\mathbb{R}^D$. To model the data, we divide $\boldsymbol{X}$ into two subsets: $\boldsymbol{X}_{in}$ containing the inliers and $\boldsymbol{X}_{out}$ containing the outliers. We assume that the inliers can be approximated by some low-dimensional structure. We are interested in the fully unsupervised setting, where we never have access to clean samples. Our goal is to identify the inliers and outliers based on the observed samples (inductive learning) and to detect newly arriving outliers (transductive learning).

We propose a regularized AE that can simultaneously learn a low-dimensional data representation and identify the outliers. Our goal is to attenuate the influence of outliers during the training of the AE, which will result in a more reliable AE mapping. To achieve this goal, we define an indicator vector $\boldsymbol{b} \in \{0, 1\}^N$ whose value $i$ indicates if the sample $\boldsymbol{x}_i$ is an inlier ($b[i] = 1$) or an outlier ($b[i] = 0$). To learn the parameters of the encoder-decoder pair ($\boldsymbol{\rho}()$ and $\boldsymbol{\psi}()$) while simultaneously identifying the inliers and outliers, we propose the following robust AE loss

$$L_d(\boldsymbol{\psi}, \boldsymbol{\rho}, \boldsymbol{b}) = \sum_i b[i]\big\|\boldsymbol{x}_i - \widehat{\boldsymbol{x}}_i\big\|_2^2 - \lambda\|\boldsymbol{b}\|_0, \qquad (1)$$

where $\widehat{\boldsymbol{x}}_i = \boldsymbol{\psi}(\boldsymbol{\rho}(\boldsymbol{x}_i))$. In equation (1), the main term is a standard AE reconstruction term computed only for samples with a value of 1 for the variable $b[i]$. The $\ell_0$ norm is used to count the number of samples included in the reconstruction error, referred to as "inliers". By balancing the error of the reconstruction and the $\ell_0$ penalty, the hyper-parameter $\lambda$ is used to control the cost associated with the number of samples used by the AE.

When the value of $\lambda$ is large, the model includes more samples. Conversely, a smaller $\lambda$ results in a sparser solution with fewer samples included by the model. If $\boldsymbol{X}_{in}$ lies near a low-dimensional manifold, the encoder-decoder pair can give a good approximation of the inliers, i.e., $\widehat{\boldsymbol{x}}_i \approx \boldsymbol{x}_i$ for $\boldsymbol{x}_i \in \boldsymbol{X}_{in}$. However, if outliers are not near the low dimensional manifold, $\|\boldsymbol{x}_i - \widehat{\boldsymbol{x}}_i\|_2^2$ is expected to be significant. Unfortunately, the $\ell_0$ norm in Eq. (1) makes the problem intractable even with a small number of samples. To overcome this limitation, we propose, following [Yamada et al., 2020, Lindenbaum et al., 2021a,b], to replace the deterministic search over the values of the indicator vector $\boldsymbol{b}$ with a probabilistic counterpart.

## 2.2 PROBABILISTIC ROBUST AUTOENCODER

We are now ready to present our probabilistic formulation for a sparse AE. Towards this goal, we multiply the samples

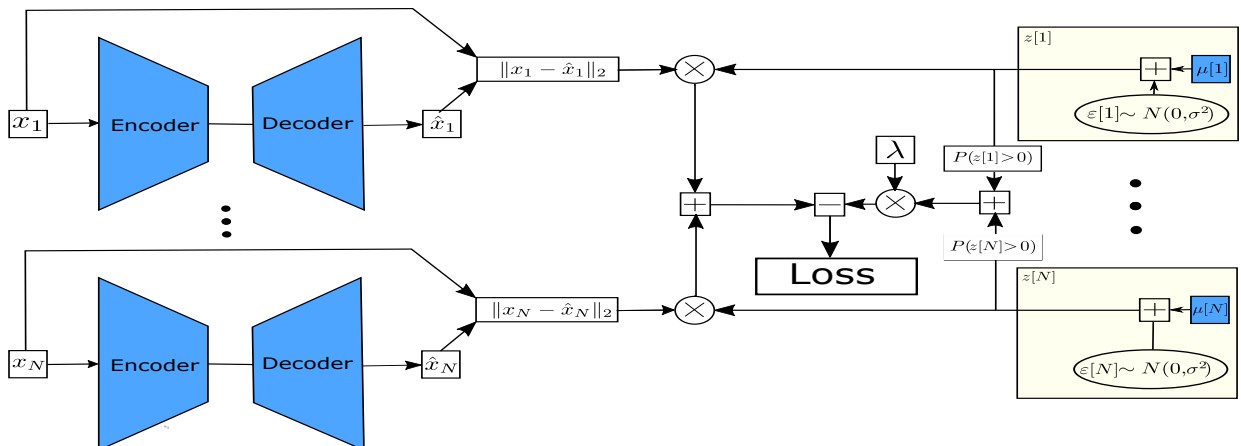

Figure 1: A schematic of the Probabilistic Robust AutoEncoder (PRAE) framework. On the left, input samples denoted by $x_i$s are multiplied by continuously relaxed Bernoulli gates. The model has an encoder, decoder, and $\mu$, all represented in blue. The network computes a reconstruction loss (in the middle) for any choice of these components. The model learns the parameters of the Bernoulli variable to select samples that are "easier" to reconstruct (as per Eq. (3)). We optimize the "blue" variables (encoder, decoder, and $\mu$) to achieve the best results. During training, outliers can be excluded, and the model learns useful mappings for detecting outliers.

by stochastic gates that relax the binary nature of the indicator vector $b$. The gates are differentiable and are designed to select a subset of samples on which the AE reconstruction error is minimized. We parameterize a stochastic gate (STG) using mean-shifted truncated Gaussian distribution. Specifically, we denote the STG random vector as $z \in [0,1]^N$, parametrized by $\mu \in \mathbb{R}^N$. Each vector entry is defined as

$$z[i] = \max(0, \min(1, \mu[i] + \epsilon[i])), \quad (2)$$

where $\epsilon[i]$ is drawn from $\mathcal{N}(0, \sigma^2)$, $\sigma$ is fixed throughout training, and $\mu[i]$ is a trainable parameter which controls the distribution of the random variable $z[i]$.

We can now incorporate the STGs into our proposed probabilistic AE loss. Formally, using a reconstruction loss, a probabilistic AE loss can be described using one of the following terms

$$L_{p_0}(\psi, \rho, \mu) = \mathbb{E}\Big( \sum_i z[i] \big\| x_i - \widehat{x}_i \big\|_2^2 - \lambda \| z \|_0 \Big), \quad (3)$$

$$L_{p_1}(\psi, \rho, \mu) = \mathbb{E}\Big( \sum_i z[i] \big\| x_i - \widehat{x}_i \big\|_2^2 - \lambda \| z \|_1 \Big), \quad (4)$$

where $\lambda$ is a regularization parameter that controls the cost associated with the number of samples included by the AE. We propose the following strategy to minimize the new loss functions (3) or (4). Given some initial guess for the encoder, and decoder, parameterized via the weights of $\psi$ and $\rho$, we draw realizations for the random vector $z$ and compute the loss value. We note that the regularization terms $\mathbb{E}(\| z \|_0) = \sum P(z[i] > 0)$, and $\mathbb{E}(\| z \|_1) = \sum \mathbb{E}(z[i])$ are parametric, and the expected value of the left term of Eqs. (3) and (4) is approximated using Monte Carlo sampling.

Then, we differentiate the loss using SGD to update the weights in $\psi$ and $\rho$, and the vector $\mu$. Figure 1 presents a schematic illustration of this procedure. For inductive learning, we use the values of the trainable vector $\mu$ as anomaly scores for each data point. A smaller value indicates that the sample should be excluded from the reconstruction loss and therefore is more anomalous. In Section 5.2, we propose two unsupervised schemes for tuning the regularization parameter $\lambda$.

## 3 RELATED WORK

Anomaly detection has been previously addressed using AEs. A straightforward approach involves using the reconstruction error of each sample to quantify if it is normal or anomalous [Sakurada and Yairi, 2014]. Since this approach does not induce regularization, the AE may overfit the outliers and learn a mapping that does not correctly characterize the normal samples. To solve this limitation, in [Chen et al., 2017], the authors propose an ensemble of AEs for anomaly detection. The idea is to train many AEs, each pruned by randomly subsampling the learned connectivities. Then, an aggregated prediction of the ensemble is used to identify the anomalies. One disadvantage of this approach is that it requires extensive computational and memory costs since it involves training hundreds of AEs. Furthermore, the proposed scheme outperforms the ensemble of AEs on several benchmark datasets (see Supplementary Material D).

Perhaps the most related method to our work is [Zhou and Paffenroth, 2017]. The authors proposed the following regularized AE objective $\| L_D - \psi(\rho(L_D)) \|_2 + \lambda \| S \|_{2,1}$ s.t. $X - L_D - S = 0$. This model aims to split the data $X$ into

two parts, $\boldsymbol{L}_D$ and $\boldsymbol{S}$ while minimizing the reconstruction error on $\boldsymbol{L}_D$. The regularization in the form of an $\ell_{2,1}$ norm is designed to sparsify the rows (samples) or columns (features) of $\boldsymbol{S}$. This way, the data is split into inliers $\boldsymbol{L}_D$ and a sparse set of outliers $\boldsymbol{S}$. To minimize the objective with the additional constraint, they use the Alternating Direction Method of Multipliers (ADMM) [Boyd et al., 2004] with an element-wise projection approach to enforce the constraint. This method differs from our approach significantly since the regularization relies on the $\ell_{2,1}$ norm applied to $\boldsymbol{S}$. The $\ell_{2,1}$ norm leads to shrinkage of values, and therefore $\boldsymbol{S}$ is not guaranteed to reflect actual samples from $\boldsymbol{X}$.

In [Ishii et al., 2020, 2022], the authors follow a similar construction to the algorithm in [Zhou and Paffenroth, 2017] but using an $\ell_0$ norm. The main difference between our method and [Zhou and Paffenroth, 2017, Ishii et al., 2020, 2022], is that they all rely on an alternating minimization procedure, in which the data is decomposed into low-rank and sparse parts. This generally leads to element-wise sparsity. Our approach differs significantly and enables simultaneous encoder-decoder pair and sample-wise sparsification learning. Furthermore, since our approach is based on gradient descent, it is directly amenable to parallel computing (using small batches). Furthermore, as demonstrated in section 5, leads to more accurate identification of outliers when applied to real and synthetic data.

## 4 ANALYSIS

The problem of robust learning with $\ell_1$ regularization was studied theoretically in the context of robust regression [Huber, 1987, Mitra et al., 2010, Yang et al., 2018]. We are interested in a related problem for unsupervised learning. Specifically, in this section, we justify the use of our proposed probabilistic AE (see Section 2.2) to solve the robust auto-encoding problem (see Section 2.1). Since the latter is not differentiable while the first is, our goal is to show that both minimization problems lead to the same solution.

First, to avoid divergence of the values of $\boldsymbol{\mu}$ in the theoretical analysis, we bound the values of $\boldsymbol{\mu}$ by

$$-M \le \boldsymbol{\mu}[i] \le M, \tag{5}$$

for some large number $M$. Note that the number $M$ is used only for the theoretical analysis and has no practical use when running the algorithm.

For any vector $\boldsymbol{b} \in \{0,1\}^N$, we define $\boldsymbol{\mu_b}$, such that $\boldsymbol{\mu_b}[i] = -M$ if $\boldsymbol{b}[i] = 0$, and $\boldsymbol{\mu_b}[i] = M$ if $\boldsymbol{b}[i] = 1$ for $i = 1 \dots N$. For any $\boldsymbol{\mu}$ we define $\boldsymbol{b}_\mu$ such that $\boldsymbol{b}_\mu[i] = sign(\boldsymbol{\mu}[i])$.

We now turn our attention to show that the deterministic optimization problem (1) (which is not differentiable) is equivalent to our probabilistic optimization (4) in the following sense

**Theorem 4.1.** *For any dataset $\boldsymbol{X}$, denote by $(\boldsymbol{\psi}_d, \boldsymbol{\rho}_d, \boldsymbol{b}_d)$ the minimizer of* (1) *and by $(\boldsymbol{\psi}_p, \boldsymbol{\rho}_p, \boldsymbol{\mu}_p)$ the minimizer of* (4). *Assume that the minimizer of* (1) *is unique and that*

$$\min_{(\boldsymbol{\psi}, \boldsymbol{\rho}, \boldsymbol{b}) \neq (\boldsymbol{\psi}_d, \boldsymbol{\rho}_d, \boldsymbol{b}_d)} L_d(\boldsymbol{\psi}, \boldsymbol{\rho}, \boldsymbol{b}) \ge L_d(\boldsymbol{\psi}_d, \boldsymbol{\rho}_d, \boldsymbol{b}_d) + \varepsilon_0 \tag{6}$$

*for some $\varepsilon_0 > 0$. Then for a sufficiently large $M > 0$ (see* (5)*), $(\boldsymbol{\psi}_d, \boldsymbol{\rho}_d) = (\boldsymbol{\psi}_p, \boldsymbol{\rho}_p)$, and for any $i = 1, \dots, L$, $b[i] = 1$ if $\mu[i] > 0$ and $b[i] = 0$ otherwise.*

In other words, if the minimizer of (1) is unique, then the encoder, decoder that minimize (1) and (4) are equivalent. Moreover, the samples included by both models (indicated by $\boldsymbol{b}$ and $\boldsymbol{z}$) are the same.

*Proof.* The proof construction is comprised of three arguments. The final argument relies on the first two and concludes the proof.

**Argument 1:** For any triplet $(\boldsymbol{\psi}_d, \boldsymbol{\rho}_d, \boldsymbol{b})$ the deterministic loss $L_d$ can be approximated by the probabilistic loss $L_{p_1}$. Namely, for any $\varepsilon, \delta > 0$ there is a value of $M > 0$ such that

$$|L_d(\boldsymbol{\psi}_d, \boldsymbol{\rho}_d, \boldsymbol{b}_d) - L_{p_1}(\boldsymbol{\psi}_d, \boldsymbol{\rho}_d, \boldsymbol{\mu_b})| \le \varepsilon,$$

with probability $1 - \delta$.

To prove this argument, we first compute the expectation $E(z)$ using definition (2), we get:

$$E(z) = \mu - \frac{1}{\sigma\sqrt{2\pi}} \int_{-\infty}^{0} t e^{-\frac{(t-\mu)^2}{2\sigma^2}} dt -$$
$$\frac{1}{\sigma\sqrt{2\pi}} \int_{1}^{\infty} t e^{-\frac{(t-\mu)^2}{2\sigma^2}} dt + \frac{1}{\sigma\sqrt{2\pi}} \int_{1}^{\infty} e^{-\frac{(t-\mu)^2}{2\sigma^2}} dt,$$

computing the integrals leads to:

$$E(z) = \frac{1}{\sqrt{2\pi}} (e^{-\frac{\mu^2}{2\sigma^2}} - e^{-\frac{(1-\mu)^2}{2\sigma^2}}) +$$
$$(\mu - 1) * \Phi(\frac{1-\mu}{\sigma}) - \mu * \Phi(-\frac{\mu}{\sigma}) + 1,$$

where $\Phi$ is the CDF of the standard normal distribution.

Since $\lim_{\mu \to \infty} E(z) = 1$, and $\lim_{\mu \to -\infty} E(z) = 0$, then, for any $\varepsilon > 0$, there is a sufficiently large $M$, such that

$$\left| \lambda \sum_i \mathbb{E}(z[i]) - \lambda \|\boldsymbol{b}\|_0 \right| < \varepsilon/2. \tag{7}$$

From the definition of $z$ we also know that for $\mu > 1$, $P(z \neq 1) = \Phi(\frac{1-\mu}{\sigma})$, and thus $\lim_{\mu \to \infty} P(z = 1) = 1$. Similarly $\lim_{\mu \to -\infty} P(z = 0) = 1$. Thus, for any $\delta$ there is $M$ large enough, such that

$$\left| \sum_i z[i] \|\boldsymbol{x}_i - \hat{\boldsymbol{x}}_i\|_2^2 - \sum_i b[i] \|\boldsymbol{x}_i - \hat{\boldsymbol{x}}_i\|_2^2 \right| < \varepsilon/2, \tag{8}$$

with probability $1 - \delta$.

Combining (7) and (8), we have that for any $\delta > 0$, there is a value of $M$ such that

$$|L_d(\boldsymbol{\psi}_d, \boldsymbol{\rho}_d, b_d) - L_{p_1}(\boldsymbol{\psi}_d, \boldsymbol{\rho}_d, \boldsymbol{\mu_b})| \leq \varepsilon,$$

with probability $1 - \delta$. This concludes the proof of Argument 1.

**Argument 2:** For any AE $(\boldsymbol{\psi}, \boldsymbol{\rho})$, the minimum $\min_{\boldsymbol{\mu}} L_{p_1}(\boldsymbol{\psi}, \boldsymbol{\rho}, \boldsymbol{\mu})$ is achieved when $\boldsymbol{\mu}[i]$ equals to either $M$ or $-M$ for all $i$.

Assume by contradiction that the minimum of $L_{p_1}$ is achieved at a point where for some $k$, $\boldsymbol{\mu}[k]$ is not either $M$ or $-M$. If $\|\boldsymbol{x}_i - \widehat{\boldsymbol{x}}_i\|_2^2 \geq \lambda$, then for $\hat{\boldsymbol{\mu}}$ such that $\hat{\boldsymbol{\mu}}[i] = \boldsymbol{\mu}[i]$ for all $i \neq k$ and $\hat{\boldsymbol{\mu}}[k] = -M$ we have that $L_{p_1}(\boldsymbol{\psi}, \boldsymbol{\rho}, \hat{\boldsymbol{\mu}}) \leq L_{p_1}(\boldsymbol{\psi}, \boldsymbol{\rho}, \boldsymbol{\mu})$, which contradicts the minimality of $L_{p_1}(\boldsymbol{\psi}, \boldsymbol{\rho}, \boldsymbol{\mu})$. In case $\|\boldsymbol{x}_i - \widehat{\boldsymbol{x}}_i\|_2^2 \leq \lambda$ a similar argument will lead to a contradiction as well.

**Argument 3:** Assume by contradiction that the minimizers of (1) and (4) are not equivalent, i.e.

$$(\boldsymbol{\psi}_d, \boldsymbol{\rho}_d, \boldsymbol{\mu}_{b_d}) \neq (\boldsymbol{\psi}_p, \boldsymbol{\rho}_p, \boldsymbol{\mu}_p). \tag{9}$$

From Argument 2 we have that $\mu_p[i] = M$ or $-M$ for all $i$. From Argument 1, we have that

$$\begin{aligned}\|L_d(\boldsymbol{\psi}_p, \boldsymbol{\rho}_p, \boldsymbol{b}_{\mu_p}) - L_{p_1}(\boldsymbol{\psi}_p, \boldsymbol{\rho}_p, \mu_p)\| \leq \varepsilon, \\ \|L_d(\boldsymbol{\psi}_d, \boldsymbol{\rho}_d, \boldsymbol{b}_d) - L_{p_1}(\boldsymbol{\psi}_d, \boldsymbol{\rho}_d, \mu_{b_d})\| \leq \varepsilon.\end{aligned} \tag{10}$$

Since $\boldsymbol{\psi}_p, \boldsymbol{\rho}_p, \boldsymbol{\mu}_p$ is the minimizer of $L_{p_1}$, we have from (10) that

$$L_d(\boldsymbol{\psi}_d, \boldsymbol{\rho}_d, \boldsymbol{b}_d) \geq L_d(\boldsymbol{\psi}_p, \boldsymbol{\rho}_p, b_{\mu_p}) - 2\varepsilon. \tag{11}$$

From Eq. (9) and the assumption of the theorem in Eq. (6), we have that

$$\|L_d(\boldsymbol{\psi}_p, \boldsymbol{\rho}_p, \boldsymbol{b}_{\mu_p}) - L_d(\boldsymbol{\psi}_d, \boldsymbol{\rho}_d, \boldsymbol{b}_d)\| \geq \varepsilon_0. \tag{12}$$

For $2\varepsilon < \varepsilon_0$, Eq. (12) contradicts Eq. (11). $\qquad\square$

## 5 EXPERIMENTS

### 5.1 ROBUST SUBSPACE RECOVERY PROBLEM

First, we test the performance of the proposed algorithms in the linear setting. While this regime has fewer applications, it is well-studied and easy to evaluate by comparing different methods. In the linear regime, the outlier detection problem is tightly related to the Robust Subspace Recovery problem (RSR). Thus, we compare our proposed scheme to baselines designed to solve the RSR problem. The RSR problem involves finding a low-dimensional (linear) subspace in a corrupted, potentially high-dimensional dataset.

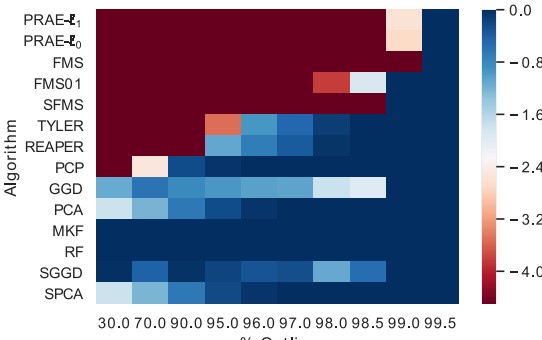

Figure 2: Comparing PRAE to several Robust Subspace Recovery (RSR) algorithms. The $y$-axis represents the different algorithms, and the $x$-axis represents different percentiles of outliers. Each box is colored according to the mean over 10 runs of the $\log()$ of the angle between the recovered subspace and the ground truth. Lower values (darker red) indicate that the subspace recovery is more accurate. The proposed PRAE can accurately recover the low dimensional subspace even in the presence of a high percentage of outliers.

For a complete overview of RSR, we refer the reader to [Lerman and Maunu, 2018b].

Following Lerman and Maunu [2018b], for any chosen percentile of outliers $r = N_{out}/N$, we generate $N = 10000$ points in $\mathbb{R}^{200}$ in the following way: first we randomly generate $\boldsymbol{X}_{in}^{low}$, a set of $N_{in} = (1 - r)N$ random points in $\mathbb{R}^{10}$. Next we generate a random linear transformation $\boldsymbol{T} \in \mathbb{R}^{200 \times 10}$, and set $\boldsymbol{X}_{in}^{high} = \boldsymbol{T}\boldsymbol{X}_{in}^{low}$. Finally, we generate $\boldsymbol{X}_{out}$ as $N_{out} = rN$ random points in $\mathbb{R}^{200}$, and define the dataset $\boldsymbol{X} = \boldsymbol{X}_{in}^{high} \cup \boldsymbol{X}_{out}$. The task is to recover $\boldsymbol{T}$ and $\boldsymbol{X}_{in}$ given the data $\boldsymbol{X}$. The accuracy is measured by the $\log$ of the angle between the recovered $\boldsymbol{T}$ and the correct $\boldsymbol{T}$ (therefore, smaller values indicate a more accurate recovery). Each experiment was performed 10 times; the outcome is the average of the 10 runs. We refer the reader to Supplementary Material A for complete implementation details.

The comparison to other algorithms under a high percentile of outliers ($r = N_{out}/N$) appears in Figure 2. We compared to leading schemes from the evaluation in [Lerman and Maunu, 2018b], namely: fast median subspace (FMS) [Lerman and Maunu, 2018a], Tyler's M-estimator (TYLER) [Zhang, 2016], REAPER [Lerman et al., 2015], the augmented Lagrange multiplier method (PCP) [Lin et al., 2010], geodesic gradient descent (GGD) [Maunu et al., 2019], and principal component analysis (PCA).

While our approach is not explicitly designed for the RSR problem, it is easy to see that our algorithms perform on par with state-of-the-art methods for RSR. Even for 99% outliers, in 7 out of 10 runs, our algorithms found exactly

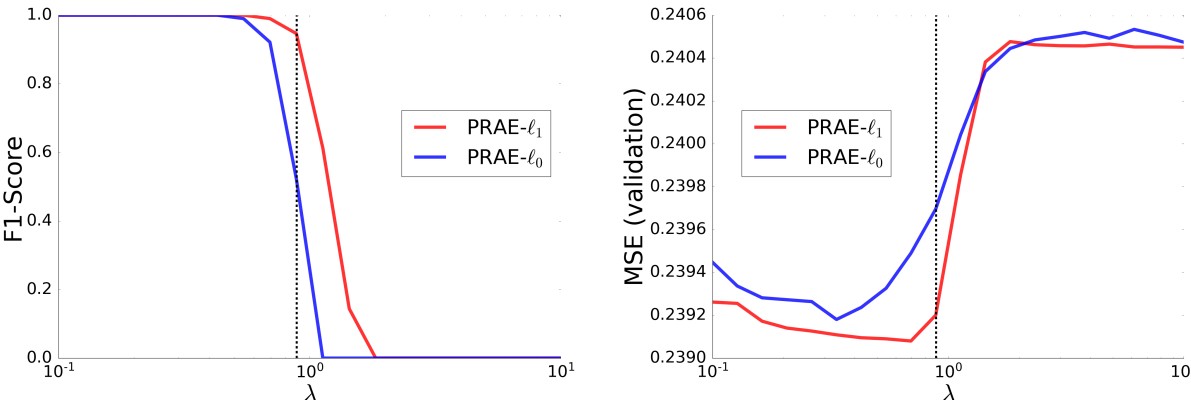

Figure 3: Phase transition of PRAE. As we increase $\lambda$ above a certain threshold, PRAE starts to include outliers, resulting in a lower F1-score (left panel) and larger reconstruction error (MSE) on unseen samples (right panel). The dashed line indicates our (unsupervised) estimation of the value of $\lambda$ in which the proposed scheme transitions from removal to the inclusion of outliers.

all the inliers. Since our approach is not designed for RSR and is focused on the more general non-linear setting, FMS recovers a more accurate subspace and requires a shorter training time. Nonetheless, this experiment highlights that our model is relatively robust to the number of outliers. We observe that PRAE can correctly recover inliers in a noisy setting. Precisely, when we use $\boldsymbol{X}_n = \boldsymbol{X} + \eta$, where the noise $\eta \sim N(0, 10^{-2}I)$ and 99% outliers, our model is still able to correctly identify a subset of inliers that are sufficient for subspace recovery in 7 out of 10 cases. However, we omit this noisy RSR experiment since it requires a more involved method for reconstructing the subspace based on the (correctly) retrieved noisy inliers.

## 5.2 UNSUPERVISED SCHEMES FOR TUNING THE REGULARIZATION PARAMETER

One practical consideration in PRAE is the choice of regularization parameter $\lambda$. In this section, we empirically study the effect of this parameter and propose two unsupervised schemes for tuning $\lambda$. We use synthetic data to demonstrate that our estimated value of $\lambda$ leads to the accurate identification of inliers and removal of all outliers.

We focus on the linear data model described in Section 5.1, but with $N = 200$, $\boldsymbol{X}_{in}^{low} \in \mathbb{R}^{150 \times 2}$, and $\boldsymbol{T} \in \mathbb{R}^{100 \times 2}$. We generate data from this model and run PRAE-$\ell_0$ and PRAE-$\ell_1$ for various values of $\lambda$ in the range $[0.1, 10]$. We run each model 20 time and record the average F1-score, computed based on precision and recall of outlier identification. Specifically, we define an outlier $\boldsymbol{x}_i$ as a sample such that after training $\mu[i] < thresh$, and an inlier otherwise. Here, $\mu[i]$ is computed based on Eq. (2) but without the injected Gaussian noise. We set $thresh$ to 0.1, although other values within $(0, 1)$ yield similar results.

In both proposed loss functions (see (3) and (4)), $\lambda$ balances

between the number of samples included by the model and the reconstruction loss. For a very large $\lambda$, we expect the model to include all samples since the regularization term would be larger than the reconstruction of $\boldsymbol{x}_i$ (for inliers and outliers). On the other hand, if $\lambda = 0$, all samples should be excluded by the model. For small values of $\lambda > 0$, we expect the model to include the inliers (since we can obtain zero reconstruction loss) and exclude the outliers. Based on the linear model experiment (described above), we observe a "phase transition" in the behavior of PRAE as a function of $\lambda$. Namely, as evident in the left panel of Figure 3 for small values of $\lambda$, PRAE accurately removes all outliers and includes all inliers.

In this example, since all samples have roughly the same energy ($\ell_2$ norm), we can propose a simple scheme for estimating the $\lambda$ value in which the phase transition occurs. Specifically, we can compute the mean energy of all samples, namely $ME = \frac{1}{N} \sum_i^N \|\boldsymbol{x}_i\|_2^2$. Since we can not reliably reconstruct the outliers (based on our data model), we expect the error for reconstructing outliers to be $\sim ME$. Therefore, for any $\lambda > ME$, PRAE-$\ell_1$ should include outliers since $\|\boldsymbol{x}_i - \hat{\boldsymbol{x}}_i\|_2^2$ is compared to $\lambda$ in loss (see (4)). On the other hand, if $\lambda < ME$, PRAE-$\ell_1$ should exclude outliers (based on the same argument). For PRAE-$\ell_0$, this argument is not precise; nonetheless, we observe that $ME$ lines well with the phase transition of both models. This is presented as a dashed black line in Figure 3.

Another scheme for tuning $\lambda$ involves finding the value that minimizes the reconstruction loss of unseen samples (a validation set). Here, we assume that inliers can be represented by a low-dimensional space while outliers can not. By evaluating the model's reconstruction error on unseen samples, we check if the model suffered from overfitting on anomalies or has used only inliers. We repeat the experiment above but evaluate the average reconstruction loss

| $\sigma_N^2$ | COPOD | kNN | ECOD | IForest | LSCP | HBOS | NTL | GOAD | RSR-AE | $\ell_{2,1}$-AE | PRAE-$\ell_0$ | PRAE-$\ell_1$ |
|---|---|---|---|---|---|---|---|---|---|---|---|---|
| 0.1 | 58.15 | 8.48 | 41.29 | 26.08 | 67.72 | 60.65 | 99.50 | 47.77 | 93.12 | 68.47 | **96.23** | **97.48** |
| 1 | 56.65 | 98.79 | 53.19 | 55.20 | 67.99 | 81.34 | 68.59 | 79.68 | 82.25 | 46.23 | **99.97** | 99.94 |
| 10 | 88.47 | 99.12 | 93.32 | 97.87 | 99.65 | 94.34 | 25.66 | 67.01 | 75.13 | 80.59 | **99.55** | 99.42 |

Table 1: Performance comparison on a synthetic dataset. Each value corresponds to the median AUC of the ROC curve of a different algorithm (column) under several anomaly variances (row). Blue and bold indicate first and second-ranked methods, respectively.

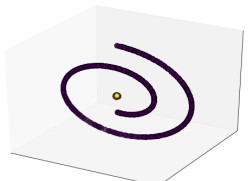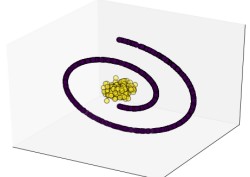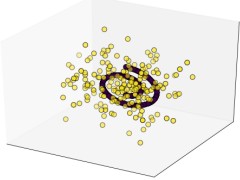

Figure 4: Examples of the synthetic narrow Swiss roll (blue) with Gaussian outliers (yellow), generated using different values of $\sigma^2 \in 0.1, 1, 10$.

on 200 unseen samples generated from the same model. As evident in the right panel of Figure 3, both models lead to the smallest reconstruction loss for $\lambda$ values that coincide with the perfect F1-score (supporting the validity of the proposed tuning scheme). We observe that PRAE-$\ell_0$ leads to a higher reconstruction error for large values of $\lambda$. This might indicate that the inclusion of all samples occurs earlier in training, leading to stronger overfitting.

# 6 OUTLIER DETECTION WITH SYNTHETIC DATA

In the synthetic example described in Section 6.3, we consider a "narrow swiss-roll", with 1000 points uniformly sampled from $[3\pi/2, 9\pi/2] \times [0, 0.1]$, and embedded into $\mathbb{R}^3$ using $(t, h) \to (t\cos(t), h, t\sin(t))$. Then, we generate additional 200 "outliers" sampled from $N(0, \sigma^2 I_3)$, where $I_3 \in \mathbb{R}^{3\times 3}$ is the identity matrix. In Figure 4 we present examples of Swiss rolls with anomalies generated using different values of $\sigma$. In this example, we use a NN with hidden layers of size $512, 256, 128, 64, 32$, and the latent space has two neurons. Since the energy of the data varies substantially across samples, we used a normalized reconstruction loss for training all AE-based methods. Specifically, we normalize the reconstruction error of each sample by the $\ell_2$ norm of the sample.

To illustrate the qualities of our scheme in a nonlinear setting, we use an example suggested by [Lai et al., 2019]. For the normal samples, we consider a "narrow swiss-roll", with $10^3$ points uniformly sampled from $[3\pi/2, 9\pi/2] \times [0, 0.1]$, and embedded into $\mathbb{R}^3$ using $(t, h) \to (t\cos(t), h, t\sin(t))$. Then, we generate additional 200 outliers sampled from $N(0, \sigma_N^2 I_3)$, where $I_3 \in \mathbb{R}^{3\times 3}$ is the identity matrix. We

generate such data with several values of $\sigma_N^2$ (see Figure 4 in Section 6), and following [Chen et al., 2017, Ishii and Takanashi, 2019], we evaluate the quality of the different methods using Receiver Operating Characteristic (ROC) curves. The ROC measures the trade-off between true positive and false positive rates. The true positive rate is defined as the ratio between identified anomalies and true anomalies, while the false positive rate is the portion of normal samples identified as anomalies. The ROC curves are summarized by measuring the AUC (area under the curve).

## 6.1 OUTLIER DETECTION ON REAL DATA

We evaluate the proposed approach on a diverse set of 41 real-world datasets with dimensions up to $32K$ and sample sizes up to $620K$. All properties appear in the right column of Table 3 in the Supplementary Material. To evaluate performance, we randomly split each dataset and use one half for training and transductive learning, and the remaining hold-out-set is used to assess the accuracy of inductive learning. The anomaly score for $x_i$ is based on the reconstruction error $\|x_i - \hat{x}_i\|_2^2$.

We compare our method to several strong baselines with code available at [Zhao et al., 2019b]. Specifically, we use kNN [Angiulli and Pizzuti, 2002], and IForest [Liu et al., 2008], which are classic methods that rely on the geometry of the data. We also use density models such as HBOS [Goldstein and Dengel, 2012], COPOD [Li et al., 2020], and ECOD [Li et al., 2022]. Finally, we compare several NN-based methods, such as $\ell_{2,1} - AE$ [Zhou and Paffenroth, 2017], RSR-AE [Lai et al., 2019], NTL [Qiu et al., 2021], and GOAD [Bergman and Hoshen, 2020].

Our model is trained using an encoder-decoder pair con-

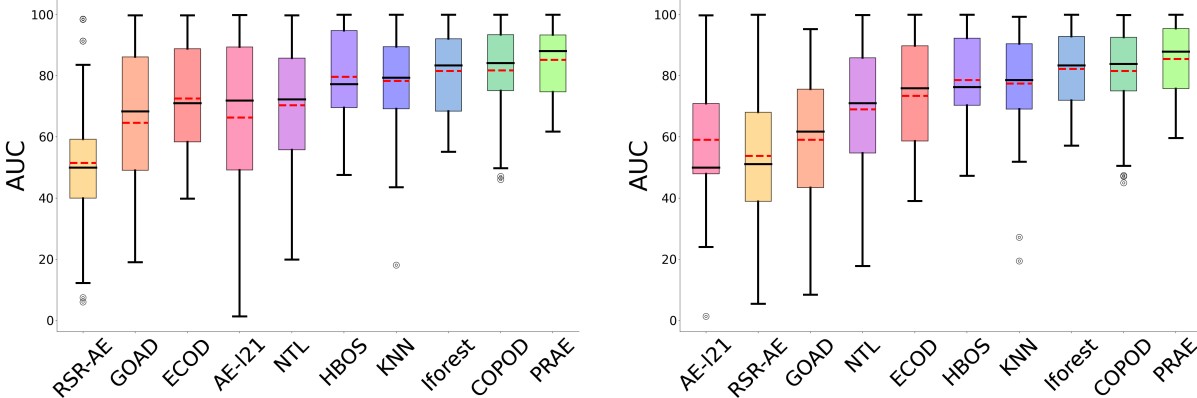

Figure 5: Outlier detection benchmark using 41 real tabular datasets. We present the box plots comparing the median AUC results of each method when applying 10 random splits of each dataset. Black lines indicate medians while red lines are the mean. Left panel: results for the inductive learning setting. Right panel: results for the transductive learning setting. The proposed PRAE outperforms all baselines in both settings. We only present here PRAE-$\ell_1$ since the $\ell_0$ led to similar results (see complete tables in the Supplementary Material).

sisting of five hidden layers, each with a size of 100. The bottleneck dimension is set to 1, but other values produce similar results (as shown in Figure 9). We use a single value of $\lambda$ equal to 1 in all our experiments, based on the results of Section 5.2. We also evaluated the dynamic tuning of $\lambda$ proposed in Section 5.2, and its results are presented in Figure 5.2. To ensure the accuracy of our results, we run all methods ten times on random splits and record the ROC for each run. In the left panel of Figure 5, we present a box plot comparing the AUC of the proposed method and all baselines evaluated for inductive learning (referred to as the "out-of-sample" setting) on all datasets. The results show that the proposed approach outperforms leading methods on many datasets. In the right panel of Figure 5, we present the AUC results on the same datasets in the transductive learning setting.

The results obtained from our proposed approach indicate that it performs well in both settings, suggesting that it can be used as a practical method for curating datasets and detecting outliers online. We believe that the reason for this success is the ability of our method to remove outliers during training. It is noteworthy that we use a single architecture and one value of $\lambda$ across all datasets, which indicates that our model doesn't require data-specific hyperparameter tuning. On the other hand, GOAD and NTL do not perform well in our setting as they were designed for training on a clean set of samples, which is not fully unsupervised. In the Supplementary Material, we have provided complete details of these evaluations, including F1 scores, and you can refer to Table 4 and 3 for more information.

In the Supplementary Material, we provide box plots indicating the stability of our approach for different initializations (Sec. B). We also demonstrate that the method is not very sensitive to hyperparameter choice D and scales well to

large datasets (Sec. E). Specifically, training on the Donor data ($600K$ samples) requires less than a minute, which is two orders of magnitude faster than GAOD and NTL. In terms of inference time, our method is faster than most baselines and requires less than $0.5$ a second for prediction on the Donor data. Furthermore, we provide reproducibility required technical details (Sections C and D). Finally, we present a deeper analysis of our results on MNISTv2 and Fashion MNIST (Sec. F).

# 7 SENSITIVITY TO HYPERPARAMETERS

In the following experiment, we evaluate the method's sensitivity to hyperparameters. Towards this goal, we run PRAE on the Musk datasets for various values of $\lambda$ and different learning rates. This dataset was arbitrarily chosen, and the results for an additional dataset ('Yeast') appear in the Supplementary Material, in Figure 8. For each value of the parameters, we repeat the experiment ten times and report the median AUC. In Figure 6, we present heatmaps with the median AUC values for each set of parameters used in the evaluation. This figure demonstrates that PRAE is relatively stable to both of these hyperparameters, and the overall AUC varies by less than $10\%$ across all evaluated values of $\lambda$ and the learning rate. As expected, if the value of $\lambda$ is small, the model's performance deteriorates. This could happen if the model removes too many inliers from the objective at an early stage of training, thus "hurting" the ability to distinguish between inliers and outliers. Another hint for the stability of the algorithm to the hyperparameter $\lambda$ is the fact that we used the same $\lambda$ for all experiments and did not choose a different value for each experiment.

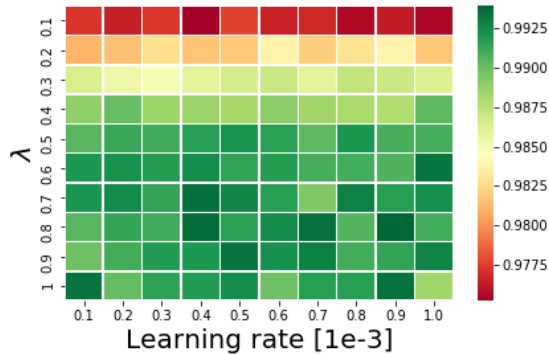

Figure 6: Heatmap presenting the AUC of the proposed approach on for several values of $\lambda$ and the learning rate using the 'Musk' data.

# 8 ABLATION STUDY

In this section, we perform an ablation study to evaluate the influence of each element of our algorithm. We focus on the 'Yeast' and 'Musk' datasets and compare PRAE-$\ell_0$, and PRAE-$\ell_1$, to the following variants: (i) **AE**- a standard AE with no regularization, anomalies are identified using the reconstruction error. (ii) **DRAE-$\ell_1$**- a variant of PRAE, but with deterministic gate values and a standard $\ell_1$ regularizer. (iii) **PRAE** ($\lambda = 0$)- a variant of PRAE, but with a regularization term= 0.

As demonstrated in Table 2 the proposed probabilistic regularization leads to improved identification of outliers compared with all variants of the method. These empirical results suggest that removing outliers throughout training with the stochastic gates can lead to more reliable identification of outliers using AEs.

| dataset | AE | DRAE-$\ell_1$ | PRAE ($\lambda = 0$) | PRAE-$\ell_0$ | PRAE-$\ell_1$ |
|---|---|---|---|---|---|
| Yeast | 74.05 | 77.26 | 77.18 | 83.37 | 83.95 |
| Musk | 95.61 | 96.81 | 88.63 | 99.17 | 98.61 |

Table 2: Ablation study. Comparing the proposed schemes PRAE-$\ell_0$, and PRAE-$\ell_1$, to three other variants. We use the same architecture and optimizer, and compare to AE, DRAE-$\ell_1$ and PRAE ($\lambda = 0$) all explained above.

# 9 STABILITY

To evaluate the stability of the proposed approach, we run the method with 10 random initialization on each of the real datasets. In Figure 5, we present box plots indicating the mean, median, and $25/75$ percentiles of PRAE-$\ell_0$ and PRAE-$\ell_1$ (left and right panels). This figure shows that the proposed approach's interquartile range (IQR) is relatively small for larger datasets (such as Shuttle, Pen, and MNISTv1). Next, we evaluate the average precision (AP)

attained by each method (when varying the anomaly score threshold). Table 6 shows the median of this value across ten runs.

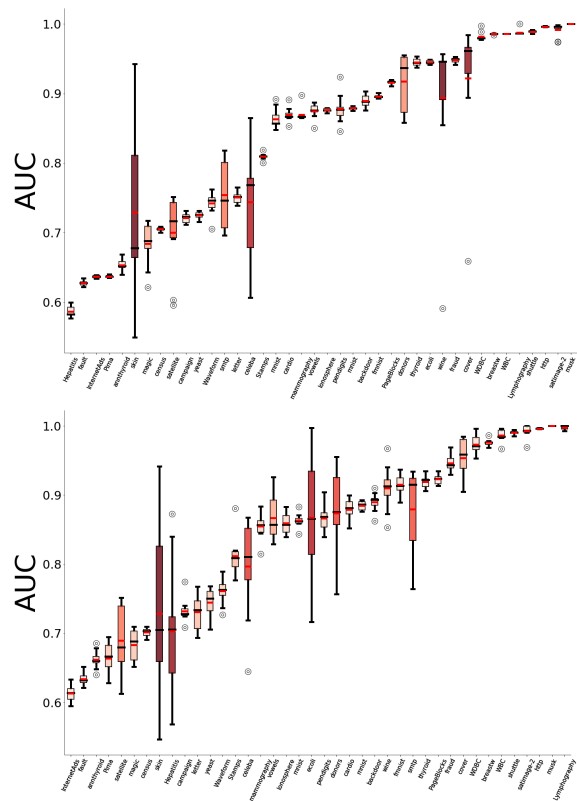

Figure 7: Box plots are presented that demonstrate the AUC of the proposed approach on real datasets. The left panel shows the variability for different random initializations of the PRAE model, whereas the right panel evaluates the variability that stems from random data splits. The color indicates the standard deviation of the AUC values.

# 10 STRENGTHS AND LIMITATIONS

The proposed method provides several advantages compared to a standard autoencoder: (1) It removes outliers along the training process and, therefore, can more accurately identify a low-dimensional subspace that represents the normal samples. (2) It provides a reliable metric for curating training data via inductive anomaly detection. (3) It learns an encoder-decoder mapping that can be used to filter anomalies from new unseen samples (transductive regime). The success of our method relies on the assumption that normal samples lie near a latent low-dimensional manifold while the outliers are diverse and do not obey such a structure. Indeed, there are cases where this assumption does not hold, and our method will not be optimal for identifying the outliers. While robust PCA assumes inliers samples are low rank, our method identifies those as samples that can be "easily"

reconstructed by an AE. To prevent the AE from overfitting to outliers, we limit its capacity using a low-dimensional bottleneck. Nonetheless, understanding the properties of an AE from an optimization perspective is an open research direction. A related problem has been studied in supervised learning in recent years [Tishby and Zaslavsky, 2015, Ronen et al., 2019]. Another limitation of our method is the regularization parameter $\lambda$. In section 5.2 we presented two heuristics for tuning $\lambda$, nonetheless, a rigorous scheme for finding the optimal value of $\lambda$ is an interesting question for future work. Another promising direction involves incorporating the proposed probabilistic robust mechanism into an attention architecture.

# 11 CONCLUSION

We have developed a framework for identifying outliers, termed Probabilistic Robust Autoencoder (PRAE). Our approach uses a regularized autoencoder to eliminate data points that are far from a low-dimensional manifold. To achieve this, we multiply each data instance with an approximately binary random variable and add a penalty term to the training process, which encourages the model to use fewer samples. We have shown that our probabilistic method is equivalent to an intractable $\ell_0$ regularized autoencoder. Our experiments demonstrate that PRAE has several advantages, including superior performance on synthetic and real datasets with varying properties such as size, dimensions, and features. This is due to the robustness of our method to contaminated training data, which makes it suitable for transductive and inductive anomaly detection.

# 12 ACKNOWLEDGMENTS

This study was supported by the National Institutes of Health (NIH) under grants R01GM131642, UM1DA051410, U54AG076043, U54AG079759, P50CA121974 and U01DA053628 (to Y.K.)

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

# Transductive and Inductive Outlier Detection with Robust Autoencoders
## (Supplementary Material)

Ofir Lindenbaum[1*]        Yariv Aizenbud[2*]        Yuval Kluger[3,4,5]

[1]Faculty of Engineering, Bar-Ilan University, Ramat Gan, Israel
[2]School of Mathematical Sciences, Tel Aviv University, Tel Aviv, Israel
[3]Program in Applied Mathematics, Yale University, New Haven, CT, USA
[4]Department of Pathology, Yale University School of Medicine, New Haven, CT, USA
[5]Computational Biology & Bioinformatics Program, Yale University, New Haven, CT, USA
[*]These authors contributed equally.

## A   ROBUST SUBSPACE RECOVERY EXPERIMENT DETAILS

Here, we describe the technical details of the experiment performed in Section 6.1. We generate $N = 10000$ points in $\mathbb{R}^{200}$ in the following way: first we generate $N_{in}$ points in $\mathbb{R}^{200}$ with distribution $N(0, I_{200})$ where $I_{200} \in \mathbb{R}^{200 \times 200}$ is the identity matrix. This set of points is orthogonally projected into a random 10 dimensional space and denoted by $\boldsymbol{X}_{in}^{high}$. The orthogonal projection is performed by projecting onto the basis of the column space of a random Gaussian matrix. $\boldsymbol{X}_{in}^{high}$ are normalized so that the expectation of the norm is 1. Next, we generate $\boldsymbol{X}_{out}$ as random points in $\mathbb{R}^{200}$ with distribution $N(0, 1/\sqrt{200} I_{200})$. This is done so that the expected value of the norm of the outliers will be equal to the expected value of the norm of the inliers. Finally, the dataset is constructed by combining $\boldsymbol{X}_{out}, \boldsymbol{X}_{in}^{high}$ and adding noise $\sim N(0, \sigma_N^2)$. For the results in Table 2 a negligible noise with $\sigma_N^2 = 10^{-8}$ was added. In the second experiment, a noise of $\sigma_N^2 = 10^{-2}$ was added.

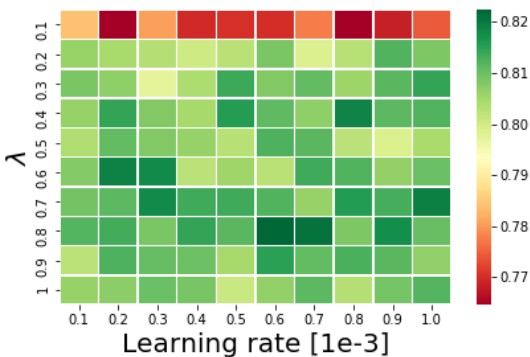

Figure 8: Heatmap presenting the AUC of the proposed approach on for several values of $\lambda$ and the learning rate using the 'Yeast' data.

## B   ADDITIONAL EXPERIMENTAL DETAILS

## C   DATASTES

All real-world datasets analyzed in this study are publicly available. Most datasets can be directly downloaded from [1]. MNIST and Fashion-MNIST can be easily obtained using Python from the "Keras" package. The Purified populations of peripheral blood monocytes (PBMCs) is a Single-cell RNA sequencing (scRNA-seq) data. It was collected by [Zheng

---

[1]http://odds.cs.stonybrook.edu/

Table 3: Inductive learning: performance comparison with several leading baselines. We present the median AUC over 10 runs. Blue and bold indicate the best and second-best methods, respectively. Additional metrics appear in the Supplementary Material.

| Dataset | kNN | IForest | ECOD | HBOS | COPOD | GOAD | NTL | RSR-AE | $\ell_{2,1}$-AE | PRAE-$\ell_0$ | PRAE-$\ell_1$ | $N$ | $D$ |
|---|---|---|---|---|---|---|---|---|---|---|---|---|---|
| Hepatitis | 79.10 | 66.71 | 76.93 | 77.22 | 81.02 | 30.25 | 99.29 | 43.65 | 66.47 | 69.10 | 70.56 | 80 | 19 |
| Wine | 43.55 | 77.54 | 58.40 | 93.49 | 88.36 | 19.00 | 90.13 | 42.62 | 64.44 | 91.08 | 90.98 | 128 | 13 |
| Lympho | 100.00 | 99.12 | 98.70 | 99.82 | 100.00 | 98.59 | 55.76 | 53.99 | 93.93 | 100.00 | 100.00 | 148 | 18 |
| WBC | 98.56 | 99.92 | 68.83 | 99.07 | 99.66 | 79.25 | 23.51 | 32.90 | 52.78 | 99.06 | 98.86 | 222 | 9 |
| Stamps | 84.57 | 86.42 | 92.36 | 90.83 | 93.43 | 49.07 | 43.47 | 48.36 | 95.45 | 84.48 | 81.66 | 340 | 9 |
| Ionos | 92.39 | 82.48 | 42.55 | 55.24 | 77.80 | 81.37 | 97.31 | 44.38 | 39.01 | 84.60 | 85.98 | 350 | 32 |
| WDBC | 96.42 | 98.00 | 99.33 | 98.93 | 99.55 | 79.44 | 66.71 | 79.54 | 98.66 | 97.70 | 96.75 | 366 | 30 |
| Breastw | 97.26 | 98.14 | 52.83 | 98.60 | 99.47 | 86.54 | 62.89 | 49.08 | 51.58 | 97.86 | 97.83 | 682 | 9 |
| Pima | 67.47 | 64.95 | 65.27 | 69.04 | 64.57 | 63.55 | 73.93 | 57.93 | 74.93 | 67.02 | 67.67 | 768 | 8 |
| YEast | 73.56 | 76.52 | 81.57 | 82.76 | 84.16 | 37.47 | 72.28 | 49.91 | 74.37 | 81.86 | 83.48 | 1,364 | 8 |
| Vowels | 95.28 | 67.72 | 99.72 | 70.33 | 50.89 | 77.91 | 54.40 | 98.43 | 99.65 | 87.71 | 87.97 | 1,456 | 12 |
| Letters | 86.35 | 58.92 | 93.29 | 57.60 | 49.79 | 37.55 | 63.96 | 48.44 | 93.22 | 72.54 | 73.15 | 1,600 | 32 |
| Cardio | 73.86 | 89.49 | 47.08 | 83.73 | 91.89 | 91.40 | 57.00 | 43.88 | 48.52 | 87.48 | 88.82 | 1,830 | 21 |
| Ecoli | 89.27 | 86.41 | 70.81 | 96.84 | 80.93 | 99.76 | 85.52 | 76.39 | 49.57 | 90.82 | 93.36 | 1831 | 21 |
| Fault | 75.15 | 55.14 | 58.15 | 47.56 | 46.03 | 70.62 | 72.59 | 37.86 | 36.69 | 64.63 | 63.34 | 1,940 | 27 |
| Internet | 58.56 | 66.23 | 84.70 | 69.62 | 68.89 | 50.80 | 51.72 | 7.42 | 78.40 | 61.40 | 61.77 | 1,966 | 1,555 |
| Musk | 87.80 | 98.77 | 60.83 | 100.00 | 94.80 | 90.47 | 54.22 | 65.20 | 72.51 | 100.00 | 100.00 | 3,062 | 166 |
| Waveform | 72.81 | 64.11 | 88.86 | 68.82 | 72.91 | 51.71 | 75.11 | 70.95 | 83.18 | 74.86 | 77.34 | 3,442 | 21 |
| Thyroid | 96.15 | 97.82 | 63.81 | 95.31 | 94.31 | 86.33 | 99.80 | 34.12 | 75.41 | 92.77 | 92.16 | 3,772 | 6 |
| MNITSv1 | 89.49 | 83.37 | 79.35 | 51.36 | 82.15 | 87.28 | 80.77 | 91.29 | 90.19 | 93.54 | 93.31 | 5127 | 784 |
| FMNIST | 88.92 | 92.12 | 83.75 | 75.64 | 88.79 | 87.98 | 85.80 | 68.25 | 89.77 | 90.32 | 89.18 | 5300 | 784 |
| PageBlock | 88.53 | 89.49 | 86.33 | 77.33 | 87.94 | 86.17 | 88.44 | 37.37 | 28.28 | 92.20 | 92.44 | 5,392 | 10 |
| Satimage | 94.53 | 99.53 | 49.18 | 97.75 | 98.10 | 75.47 | 40.93 | 48.05 | 52.42 | 99.23 | 99.41 | 5,802 | 36 |
| PBMC | 52.05 | 83.37 | 86.12 | 94.35 | 89.11 | 91.33 | 85.79 | 91.29 | 90.19 | 93.54 | 93.31 | 6,300 | 32738 |
| Satellite | 69.21 | 68.46 | 63.03 | 75.79 | 62.82 | 50.87 | 53.48 | 48.07 | 29.87 | 72.25 | 70.93 | 6,434 | 36 |
| Pendigits | 74.24 | 92.11 | 63.39 | 92.77 | 90.26 | 55.56 | 90.66 | 55.09 | 66.51 | 87.03 | 86.42 | 6,870 | 16 |
| Annthyroid | 79.41 | 81.96 | 94.60 | 62.67 | 78.32 | 69.19 | 61.79 | 5.96 | 95.10 | 65.54 | 66.73 | 7,200 | 6 |
| MNISTv2 | 85.76 | 77.18 | 63.41 | 57.79 | 77.44 | 75.34 | 50.17 | 55.58 | 71.87 | 85.96 | 86.56 | 7,602 | 78 |
| Mammo. | 84.16 | 85.30 | 91.86 | 83.06 | 90.33 | 46.73 | 75.91 | 18.83 | 1.28 | 85.71 | 85.16 | 11,182 | 6 |
| Magic | 80.25 | 72.15 | 43.73 | 71.06 | 68.02 | 64.97 | 70.54 | 51.86 | 41.16 | 70.31 | 70.77 | 19,020 | 10 |
| Campaign | 73.89 | 68.98 | 71.02 | 76.87 | 78.48 | 43.91 | 62.86 | 12.23 | 78.15 | 72.53 | 72.60 | 41,188 | 62 |
| Shuttle | 65.68 | 99.44 | 88.81 | 98.54 | 99.44 | 20.28 | 19.87 | 54.85 | 88.77 | 98.92 | 98.60 | 49,096 | 9 |
| STMP | 91.74 | 90.41 | 89.21 | 86.94 | 90.63 | 49.96 | 70.68 | 83.54 | 89.46 | 87.07 | 88.12 | 95,156 | 3 |
| Backdoor | 75.48 | 72.50 | 97.44 | 73.72 | 79.01 | 89.97 | 92.66 | 52.62 | 57.76 | 89.59 | 88.80 | 95,328 | 193 |
| CelebA | 63.96 | 66.36 | 44.05 | 75.17 | 75.10 | 27.81 | 64.98 | 53.04 | 47.23 | 77.41 | 74.81 | 202,598 | 39 |
| Skin | 64.07 | 65.81 | 39.76 | 59.34 | 47.05 | 46.37 | 77.06 | 27.21 | 33.12 | 63.71 | 75.64 | 245,056 | 3 |
| Fraud | 95.04 | 94.69 | 75.58 | 94.77 | 94.98 | 86.15 | 94.72 | 59.23 | 73.48 | 94.86 | 95.16 | 284,806 | 29 |
| Cover | 78.35 | 87.51 | 55.40 | 71.46 | 88.41 | 34.40 | 99.03 | 22.37 | 49.23 | 95.19 | 95.28 | 286,048 | 10 |
| Census | 67.02 | 60.23 | 48.96 | 60.90 | 67.50 | 68.37 | 74.40 | 50.00 | 52.35 | 70.06 | 69.95 | 299,284 | 500 |
| Http | 18.02 | 99.94 | 60.39 | 99.43 | 99.15 | 51.04 | 92.85 | 40.00 | 44.96 | 99.59 | 99.60 | 567,498 | 3 |
| Donors | 64.25 | 80.14 | 95.69 | 73.96 | 81.52 | 58.12 | 52.00 | 98.55 | 99.89 | 85.52 | 88.71 | 619,326 | 10 |
| Median AUC | 79.41 | 83.37 | 71.02 | 77.22 | 84.16 | 68.37 | 72.28 | 49.91 | 71.87 | **87.07** | **88.12** | | |
| Median Rank | 6 | 5 | 7 | 5 | 5 | 8 | 8 | 10 | 8 | **4** | **4** | | |

et al., 2017] and contains more than $90,000$ cells with $32,738$ genes. We randomly sample $6,000$ CD34 cells as our normal samples and add 300 cells from the remaining 8 cell types. In scRNA-seq, there is typically some level of false annotations; therefore, we design this example to evaluate the capabilities of PRAE in curating a set of CD34 from contamination by other cell types.

# D  BASELINES AND HYPERPARAMETERS

In the following section, we describe all baselines and hyperparameters used to evaluate anomaly detection using real and synthetic data. We train our proposed AE with an encoder-decoder pair with five hidden layers of size 100; the latent dimension (bottleneck layer) is 1. We use the heuristic proposed in Section 6.2 to tune the regularization parameter to $\lambda = 1 < ME$. We evaluate the stability of our approach to the choice of latent dimension, and to the use of variable regularization parameter (the second tuning scheme presented in the main text). As demonstrated in Figure 9 our model performs well with different latent dimensions, and also the global regularization works as well as the datasets specific one.

Table 4: Transductive learning: performance comparison with several leading anomaly detection baselines in the "out-of-sample" setting. We present the median AUC over 10 runs. Additional metrics appear in the Supplementary Material.

| Dataset | kNN | IForest | ECOD | HBOS | COPOD | GOAD | NTL | RSR-AE | $\ell_{2,1}$-AE | PRAE-$\ell_0$ | PRAE-$\ell_1$ | $N$ | $D$ |
|---|---|---|---|---|---|---|---|---|---|---|---|---|---|
| Hepatitis | 78.86 | 73.38 | 77.00 | 72.29 | 81.11 | 8.40 | 99.30 | 43.22 | 50.00 | 67.21 | 80.08 | 80 | 19 |
| Wine | 27.19 | 81.76 | 60.92 | 91.94 | 89.63 | 8.67 | 90.94 | 59.69 | 57.26 | 88.23 | 88.47 | 128 | 13 |
| Lympho | 98.61 | 99.82 | 100.00 | 99.86 | 99.82 | 22.07 | 57.95 | 100.00 | 92.50 | 100.00 | 99.71 | 148 | 18 |
| WBC | 96.79 | 98.95 | 82.35 | 98.65 | 99.25 | 56.45 | 27.08 | 33.82 | 47.67 | 99.04 | 98.73 | 222 | 9 |
| Stamps | 88.77 | 87.63 | 93.76 | 91.06 | 92.30 | 44.84 | 39.19 | 38.21 | 94.75 | 86.11 | 85.78 | 340 | 9 |
| Ionos | 91.46 | 82.51 | 39.82 | 52.03 | 78.78 | 86.71 | 96.49 | 51.24 | 39.43 | 85.32 | 82.97 | 350 | 32 |
| WDBC | 99.33 | 98.17 | 99.25 | 99.11 | 99.66 | 43.02 | 65.82 | 79.59 | 98.80 | 96.76 | 96.13 | 366 | 30 |
| Breastw | 98.41 | 97.87 | 53.25 | 98.57 | 99.41 | 81.14 | 27.10 | 49.78 | 50.00 | 98.04 | 96.98 | 682 | 9 |
| Pima | 73.53 | 65.78 | 66.48 | 69.43 | 66.02 | 60.95 | 73.75 | 56.00 | 74.15 | 67.17 | 69.89 | 768 | 8 |
| YEast | 73.16 | 79.30 | 81.57 | 78.98 | 83.91 | 36.13 | 72.34 | 79.54 | 70.58 | 83.37 | 83.95 | 1,364 | 8 |
| Vowels | 97.80 | 73.74 | 98.60 | 66.73 | 47.40 | 95.23 | 52.42 | 97.87 | 99.53 | 87.38 | 87.89 | 1,456 | 12 |
| Letters | 87.79 | 62.27 | 92.29 | 58.13 | 50.50 | 90.46 | 62.03 | 38.98 | 50.00 | 71.93 | 68.83 | 1,600 | 32 |
| Cardio | 73.72 | 89.39 | 46.45 | 83.12 | 92.57 | 37.60 | 56.50 | 41.80 | 47.98 | 88.31 | 89.29 | 1,830 | 21 |
| Ecoli | 90.51 | 86.41 | 70.81 | 81.44 | 80.97 | 42.68 | 71.04 | 76.80 | 44.21 | 88.94 | 89.09 | 1831 | 21 |
| Fault | 71.46 | 57.10 | 58.64 | 47.31 | 45.01 | 74.50 | 72.70 | 37.43 | 36.46 | 62.55 | 63.65 | 1,940 | 27 |
| Internet | 60.49 | 65.92 | 84.53 | 69.21 | 66.41 | 33.38 | 54.32 | 6.46 | 50.00 | 62.14 | 59.62 | 1,966 | 1,555 |
| Musk | 69.12 | 98.51 | 57.04 | 100.00 | 94.34 | 93.21 | 53.91 | 63.92 | 70.91 | 100.00 | 100.00 | 3,062 | 166 |
| Waveform | 72.94 | 62.56 | 88.81 | 70.62 | 74.08 | 69.62 | 75.14 | 71.18 | 50.00 | 74.40 | 75.09 | 3,442 | 21 |
| Thyroid | 96.13 | 97.78 | 71.57 | 95.54 | 93.89 | 72.15 | 99.89 | 57.52 | 68.91 | 91.28 | 89.43 | 3,772 | 6 |
| MNITSv1 | 89.49 | 83.37 | 79.35 | 77.09 | 80.42 | 79.64 | 78.32 | 91.29 | 90.39 | 93.54 | 93.31 | 5127 | 784 |
| FMNIST | 89.24 | 91.79 | 83.75 | 74.75 | 88.60 | 66.49 | 85.86 | 82.86 | 90.69 | 93.89 | 93.99 | 5300 | 784 |
| PageBlock | 85.36 | 88.83 | 89.79 | 76.33 | 87.27 | 75.62 | 77.83 | 45.98 | 50.00 | 92.59 | 92.96 | 5,392 | 10 |
| Satimage | 91.57 | 99.01 | 46.40 | 97.36 | 96.89 | 81.73 | 36.18 | 62.28 | 49.33 | 98.94 | 99.05 | 5,802 | 36 |
| PBMC | 51.90 | 88.71 | 85.65 | 70.36 | 88.84 | 81.76 | 85.85 | 51.10 | 64.26 | 90.91 | 91.30 | 6,300 | 32,738 |
| Satellite | 67.84 | 68.21 | 57.93 | 75.37 | 63.93 | 61.72 | 54.77 | 39.46 | 23.96 | 70.87 | 73.55 | 6,434 | 36 |
| Pendigits | 72.46 | 92.86 | 64.36 | 92.33 | 91.08 | 56.03 | 92.85 | 55.34 | 50.00 | 88.46 | 87.72 | 6,870 | 16 |
| Annthyroid | 78.60 | 81.78 | 95.40 | 62.31 | 77.02 | 67.29 | 60.61 | 5.39 | 50.00 | 66.47 | 67.02 | 7,200 | 6 |
| MNISTv2 | 83.13 | 76.15 | 66.60 | 57.93 | 76.94 | 65.78 | 50.04 | 58.22 | 65.07 | 86.16 | 84.55 | 7,602 | 78 |
| Mammo. | 85.78 | 84.83 | 92.11 | 83.18 | 90.82 | 45.81 | 76.33 | 19.10 | 1.29 | 85.82 | 84.83 | 11,182 | 6 |
| Magic | 80.84 | 72.00 | 44.99 | 71.40 | 68.30 | 64.68 | 70.37 | 50.83 | 43.02 | 70.37 | 71.58 | 19,020 | 10 |
| Campaign | 73.87 | 69.34 | 73.97 | 76.65 | 78.09 | 43.42 | 62.20 | 13.53 | 81.73 | 72.31 | 72.92 | 41,188 | 62 |
| Shuttle | 65.62 | 99.41 | 88.37 | 98.67 | 99.52 | 21.34 | 17.77 | 68.11 | 86.97 | 98.84 | 98.83 | 49,096 | 9 |
| STMP | 93.15 | 89.92 | 91.65 | 77.05 | 91.52 | 49.58 | 70.73 | 81.88 | 49.96 | 90.29 | 97.59 | 95,156 | 3 |
| Backdoor | 75.28 | 72.41 | 95.33 | 73.94 | 78.74 | 90.99 | 93.07 | 51.13 | 58.54 | 88.96 | 87.47 | 95,328 | 193 |
| CelebA | 62.77 | 65.75 | 49.04 | 74.60 | 75.04 | 28.06 | 66.66 | 49.37 | 46.46 | 77.26 | 76.75 | 202,598 | 39 |
| Skin | 61.84 | 66.15 | 39.08 | 59.27 | 47.12 | 49.74 | 83.33 | 29.52 | 29.29 | 63.46 | 75.79 | 245,056 | 3 |
| Fraud | 93.07 | 94.60 | 75.93 | 95.49 | 94.45 | 90.90 | 94.55 | 59.14 | 50.00 | 94.81 | 95.49 | 284,806 | 29 |
| Cover | 77.06 | 87.30 | 59.68 | 71.18 | 88.41 | 66.87 | 97.98 | 27.82 | 51.76 | 95.23 | 98.28 | 286,048 | 10 |
| Census | 67.66 | 60.44 | 48.80 | 60.87 | 67.31 | 68.30 | 80.13 | 50.00 | 50.00 | 70.05 | 70.34 | 299,284 | 500 |
| Http | 19.41 | 99.94 | 60.35 | 99.43 | 99.16 | 49.40 | 93.01 | 30.44 | 45.75 | 99.60 | 99.61 | 567,498 | 3 |
| Donors | 63.50 | 80.09 | 96.89 | 74.02 | 81.57 | 58.58 | 54.47 | 99.21 | 99.78 | 85.35 | 88.68 | 619,326 | 10 |
| Median AUC | 78.60 | 83.37 | 75.93 | 76.33 | 83.91 | 61.72 | 71.04 | 51.13 | 50.00 | **88.23** | 87.89 | | |
| Median rank | 5 | 5 | 7 | 5 | 5 | 8 | 7 | 10 | 8 | **4** | 3 | | |

We use Adam optimizer with a learning rate of $N \cdot 10^{-6}$ where $N$ is the number of samples in the dataset. Our batch size is $N/10$.

**Isolation Forest (IForest) [Liu et al., 2008]** seeks the minimum number of splits required to isolate each sample. Then, an ensemble of trees is used to define an anomaly score.

**Histogram-based Outlier Score (HBOS) [Goldstein and Dengel, 2012]** is a probabilistic approach that creates histograms to identify anomalies as samples with low density.

**Deep One-Class Classification (Deep-SVDD) [Ruff et al., 2018]** extends OC-SVM by introducing a NN to model the decision boundary of the normal data. Here, we evaluated several DSVDD architectures, but none led to results that are competitive with the baselines presented in the main text.

**Robust deep autoencoders ($\ell_{2,1} - AE$) [Zhou and Paffenroth, 2017]** uses an $\ell_{2,1}$ to regularize an AE that attempts to reconstruct the data while removing outliers. The approach is explained in the main text. We use an AE with the same architecture as the proposed approach in all examples.

**Ensemble of Autoencoders (RandNet) [Chen et al., 2017]** The method relies on an aggregated ensemble of AEs. Our method outperforms the results presented by the authors on the datasets reported in our main text.

Table 5: Transductive learning: performance comparison with several leading baselines. We present the median average precision (AP) over 10 runs. Blue and bold indicate the best and second-best methods, respectively.

| Dataset | kNN | IForest | ECOD | HBOS | COPOD | GOAD | NTL | RSR-AE | $\ell_{2,1}$-AE | PRAE-$\ell_0$ | PRAE-$\ell_1$ | $N$ | $D$ |
|---|---|---|---|---|---|---|---|---|---|---|---|---|---|
| Hepatitis | 38.23 | 27.69 | 24.53 | 35.27 | 45.29 | 13.40 | 22.82 | 7.33 | 11.22 | 28.12 | 29.08 | 80 | 19 |
| Wine | 30.37 | 20.57 | 10.94 | 36.38 | 45.00 | 5.52 | 76.86 | 7.42 | 3.02 | 39.81 | 37.04 | 128 | 13 |
| Lympho | 95.83 | 100.00 | 80.56 | 90.48 | 100.00 | 22.95 | 13.93 | 51.49 | 53.33 | 100.00 | 100.00 | 148 | 18 |
| WBC | 85.14 | 94.62 | 29.62 | 76.68 | 92.36 | 2.74 | 3.19 | 15.14 | 11.08 | 81.48 | 86.22 | 222 | 9 |
| Stamps | 28.89 | 32.54 | 55.29 | 33.53 | 39.77 | 7.68 | 17.29 | 29.13 | 27.60 | 31.53 | 23.76 | 340 | 9 |
| Ionos | 92.19 | 77.68 | 10.44 | 34.77 | 66.97 | 51.33 | 82.90 | 11.73 | 10.96 | 82.52 | 82.96 | 350 | 32 |
| WDBC | 58.20 | 67.87 | 86.19 | 71.19 | 82.20 | 1.68 | 78.94 | 34.50 | 8.05 | 70.01 | 58.33 | 366 | 30 |
| Breastw | 93.98 | 93.34 | 3.42 | 95.35 | 99.07 | 70.89 | 26.89 | 3.02 | 3.05 | 94.21 | 94.11 | 682 | 9 |
| Pima | 50.13 | 51.93 | 43.47 | 57.76 | 50.81 | 32.74 | 38.32 | 54.53 | 61.98 | 49.65 | 48.51 | 768 | 8 |
| YEast | 13.52 | 19.00 | 20.22 | 20.81 | 39.54 | 4.58 | 12.13 | 7.51 | 4.99 | 15.66 | 16.97 | 1364 | 8 |
| Vowels | 48.18 | 11.58 | 73.14 | 8.63 | 3.48 | 43.67 | 6.19 | 95.69 | 99.45 | 22.27 | 23.23 | 1,456 | 12 |
| Letters | 27.84 | 8.66 | 23.85 | 8.36 | 7.35 | 31.48 | 10.56 | 2.72 | 2.10 | 17.17 | 16.49 | 1,600 | 32 |
| Cardio | 38.77 | 53.30 | 32.44 | 47.29 | 58.44 | 14.28 | 5.83 | 30.60 | 35.98 | 43.25 | 45.26 | 1,830 | 21 |
| Ecoli | 40.71 | 55.29 | 14.47 | 30.74 | 25.05 | 12.10 | 11.37 | 21.70 | 4.76 | 74.72 | 71.61 | 1831 | 21 |
| Fault | 53.66 | 41.05 | 52.17 | 35.98 | 31.69 | 50.09 | 43.33 | 21.34 | 23.33 | 47.02 | 46.39 | 1,940 | 27 |
| Internet | 26.87 | 37.52 | 7.28 | 53.28 | 51.14 | 15.11 | 14.77 | 1.32 | 2.39 | 30.05 | 30.96 | 1,966 | 1,555 |
| Musk | 36.21 | 69.30 | 42.54 | 99.89 | 38.00 | 63.54 | 13.62 | 48.99 | 53.31 | 100.00 | 100.00 | 3,062 | 166 |
| Waveform | 10.30 | 5.19 | 10.79 | 5.23 | 6.18 | 6.45 | 9.68 | 8.40 | 5.92 | 16.39 | 18.41 | 3,442 | 21 |
| Thyroid | 35.87 | 55.70 | 17.02 | 52.83 | 18.33 | 23.07 | 18.11 | 3.44 | 3.74 | 33.55 | 37.06 | 3,772 | 6 |
| MNITSv1 | 26.28 | 28.58 | 15.59 | 31.87 | 24.67 | 12.97 | 15.23 | 2.83 | 4.53 | 28.67 | 27.61 | 5127 | 784 |
| FMNIST | 52.08 | 49.90 | 38.19 | 10.66 | 39.82 | 18.37 | 23.18 | 4.04 | 5.17 | 51.84 | 49.44 | 5300 | 784 |
| PageBlock | 50.04 | 45.23 | 55.39 | 35.73 | 36.10 | 42.12 | 49.02 | 0.15 | 0.03 | 60.06 | 59.69 | 5,392 | 10 |
| Satimage | 40.40 | 87.91 | 24.46 | 76.96 | 82.24 | 13.36 | 85.04 | 28.05 | 24.21 | 93.67 | 95.58 | 5,802 | 36 |
| Satellite | 53.79 | 66.19 | 4.03 | 67.97 | 56.89 | 42.86 | 71.50 | 2.38 | 3.25 | 69.03 | 67.74 | 6,434 | 36 |
| Pendigits | 9.67 | 28.97 | 53.81 | 24.58 | 17.96 | 2.50 | 16.87 | 59.15 | 35.15 | 11.37 | 13.13 | 6,870 | 16 |
| Annthyroid | 22.38 | 30.80 | 22.12 | 23.39 | 17.15 | 16.35 | 13.37 | 0.09 | 0.17 | 19.07 | 18.40 | 7,200 | 6 |
| MNISTv2 | 39.06 | 26.67 | 38.05 | 10.96 | 21.32 | 22.76 | 24.85 | 27.97 | 42.94 | 40.37 | 41.42 | 7,602 | 78 |
| Mammo. | 18.51 | 17.19 | 12.29 | 12.95 | 40.50 | 5.40 | 3.67 | 0.61 | 0.48 | 15.87 | 17.23 | 11,182 | 6 |
| Magic | 73.50 | 62.47 | 34.75 | 61.98 | 59.20 | 51.35 | 41.59 | 34.08 | 33.73 | 64.40 | 64.58 | 19,020 | 10 |
| Campaign | 27.93 | 25.49 | 75.52 | 35.13 | 36.58 | 11.10 | 31.65 | 24.20 | 58.92 | 27.36 | 27.84 | 41,188 | 62 |
| Shuttle | 16.98 | 94.47 | 35.49 | 96.66 | 96.29 | 9.96 | 49.67 | 6.48 | 8.24 | 90.53 | 91.66 | 49,096 | 9 |
| STMP | 27.18 | 0.58 | 14.83 | 0.46 | 0.45 | 1.02 | 46.03 | 5.21 | 2.41 | 34.26 | 30.64 | 95,156 | 3 |
| Backdoor | 37.11 | 3.72 | 66.30 | 4.96 | 6.79 | 57.33 | 37.61 | 1.14 | 1.09 | 50.45 | 51.09 | 95,328 | 193 |
| CelebA | 3.92 | 4.21 | 2.67 | 9.49 | 9.43 | 1.55 | 4.28 | 1.28 | 1.45 | 9.75 | 9.30 | 202,598 | 39 |
| Skin | 23.82 | 24.90 | 4.07 | 23.42 | 17.82 | 20.69 | 30.83 | 4.99 | 3.42 | 21.80 | 21.68 | 245,056 | 3 |
| Fraud | 7.09 | 11.99 | 4.71 | 24.36 | 27.43 | 17.12 | 23.68 | 1.78 | 2.22 | 15.14 | 16.66 | 284,806 | 29 |
| Cover | 3.49 | 3.81 | 7.24 | 2.54 | 6.62 | 1.99 | 8.81 | 4.47 | 6.03 | 14.03 | 9.70 | 286,048 | 10 |
| Census | 9.23 | 7.60 | 18.15 | 7.28 | 8.74 | 8.46 | 5.72 | 74.80 | 20.83 | 9.69 | 9.62 | 299,284 | 500 |
| Http | 1.19 | 96.87 | 2.38 | 36.68 | 30.24 | 0.44 | 33.00 | 1.96 | 2.69 | 47.95 | 46.79 | 567,498 | 3 |
| Donors | 11.56 | 11.62 | 50.98 | 11.93 | 20.71 | 6.08 | 5.87 | 33.26 | 89.72 | 32.49 | 31.89 | 619,326 | 10 |
| Median AP | 33.12 | 31.67 | 24.16 | 34.15 | 36.34 | 13.84 | 20.47 | 7.47 | 5.98 | 37.03 | 37.05 | | |
| Median Rank | 5.4 | 4.5 | 7 | 5 | 6 | 8.5 | 7.5 | 9 | 10 | **4** | **4** | | |

**Copula Based Outlier Detector (COPOD) [Li et al., 2020]** is a parameter-free method that uses a copula to model the density of the data.

**Empirical Cumulative Distribution-based Outlier Detector (ECOD) [Li et al., 2022]** is another density-based method that uses the per dimension tail probabilities to assign scores for outliers.

**Classification-Based Anomaly Detection for General Data (GOAD) [Bergman and Hoshen, 2020]** is a NN classification based method for anomaly detection. We use the *kdd* architecture as termed by the authors since it was reported to demonstrate the best accuracy as evaluated by [Shenkar and Wolf, 2022].

**Neural Transformation Learning for Deep Anomaly (NTL) [Qiu et al., 2021]** another classification-based scheme that relies on learned transformation of tabular data. We adopt the *kdd* architecture as detailed by the author to lead to the best performance.

# E  RUNNING TIME AND PLATFORM DETAILS

All the experiments were run on a server with an Intel(R) Xeon(R) CPU E5-2620 v3 @ 2.40GHz CPU and one GeForce GTX 1080 GPU with 8GB of memory. Our method scales like a standard autoencoder, and every iteration requires $\mathcal{O}(M + N)$

Table 6: Inductive learning: performance comparison with several leading baselines. We present the median average precision (AP) over 10 runs. Blue and bold indicate the best and second-best methods, respectively.

| Dataset | kNN | IForest | ECOD | HBOS | COPOD | GOAD | NTL | RSR-AE | $\ell_{2,1}$-AE | PRAE-$\ell_0$ | PRAE-$\ell_1$ | $N$ | $D$ |
|---|---|---|---|---|---|---|---|---|---|---|---|---|---|
| Hepatitis | 47.30 | 32.65 | 24.53 | 38.45 | 39.38 | 17.11 | 26.17 | 7.64 | 29.76 | 27.16 | 28.89 | 80 | 19 |
| Wine | 25.92 | 32.96 | 10.94 | 31.90 | 39.65 | 8.26 | 70.24 | 8.23 | 5.19 | 32.11 | 36.48 | 128 | 13 |
| Lympho | 95.83 | 95.83 | 80.56 | 91.67 | 95.83 | 9.13 | 25.24 | 16.70 | 38.69 | 100.00 | 95.83 | 148 | 18 |
| WBC | 73.82 | 92.87 | 29.62 | 82.11 | 92.54 | 2.73 | 21.98 | 12.15 | 38.83 | 89.84 | 84.06 | 222 | 9 |
| Stamps | 29.13 | 40.65 | 55.29 | 36.69 | 39.71 | 7.66 | 30.45 | 22.76 | 62.49 | 29.30 | 34.63 | 340 | 9 |
| Ionos | 92.06 | 77.25 | 10.44 | 34.58 | 68.93 | 50.93 | 84.40 | 9.81 | 9.28 | 80.08 | 79.71 | 350 | 32 |
| WDBC | 43.22 | 58.10 | 86.19 | 86.09 | 79.28 | 1.65 | 82.86 | 36.45 | 85.44 | 51.56 | 52.76 | 366 | 30 |
| Breastw | 91.76 | 96.60 | 3.42 | 95.97 | 98.82 | 72.83 | 27.68 | 3.41 | 2.99 | 94.49 | 94.63 | 682 | 9 |
| Pima | 54.51 | 49.54 | 43.47 | 55.74 | 53.48 | 32.71 | 36.37 | 52.80 | 63.47 | 46.95 | 48.49 | 768 | 8 |
| YEast | 16.70 | 16.53 | 20.22 | 19.90 | 39.60 | 6.91 | 10.38 | 8.74 | 5.16 | 16.86 | 17.17 | 1364 | 8 |
| Vowels | 42.61 | 12.65 | 73.14 | 8.60 | 3.71 | 14.45 | 8.15 | 97.19 | 99.07 | 17.51 | 20.36 | 1,456 | 12 |
| Letters | 29.18 | 8.70 | 23.85 | 7.74 | 7.46 | 5.21 | 15.05 | 3.33 | 14.00 | 16.00 | 17.78 | 1,600 | 32 |
| Cardio | 37.97 | 48.04 | 32.44 | 45.54 | 57.65 | 54.27 | 5.64 | 25.61 | 31.07 | 46.73 | 48.47 | 1,830 | 21 |
| Ecoli | 70.83 | 44.44 | 14.47 | 32.86 | 12.88 | 18.13 | 15.04 | 35.20 | 9.09 | 79.95 | 82.23 | 1831 | 21 |
| Fault | 51.01 | 40.10 | 52.17 | 38.37 | 31.35 | 54.01 | 42.57 | 21.04 | 24.60 | 44.24 | 45.81 | 1,940 | 27 |
| Internet | 24.07 | 35.64 | 7.28 | 50.32 | 50.94 | 17.90 | 16.12 | 1.32 | 4.67 | 28.73 | 29.21 | 1,966 | 1,555 |
| Musk | 45.48 | 67.53 | 42.54 | 99.91 | 31.94 | 50.60 | 8.45 | 45.28 | 56.00 | 100.00 | 100.00 | 3,062 | 166 |
| Waveform | 13.48 | 5.01 | 10.79 | 4.80 | 5.71 | 3.88 | 8.87 | 8.39 | 15.02 | 20.42 | 21.27 | 3,442 | 21 |
| Thyroid | 28.60 | 53.26 | 17.02 | 47.60 | 17.65 | 38.83 | 23.69 | 2.91 | 6.88 | 33.94 | 32.75 | 3,772 | 6 |
| MNITSv1 | 28.80 | 29.49 | 15.59 | 31.45 | 21.67 | 24.91 | 18.94 | 2.94 | 41.24 | 28.10 | 29.22 | 5127 | 784 |
| FMNIST | 52.33 | 50.03 | 38.19 | 10.52 | 42.31 | 68.84 | 27.55 | 4.08 | 47.11 | 50.76 | 53.09 | 5300 | 784 |
| PageBlock | 50.02 | 47.02 | 55.39 | 33.40 | 38.01 | 55.39 | 45.64 | 0.30 | 0.02 | 58.97 | 58.66 | 5,392 | 10 |
| Satimage | 40.50 | 90.19 | 24.46 | 76.27 | 76.46 | 21.50 | 93.69 | 42.68 | 25.01 | 97.22 | 96.11 | 5,802 | 36 |
| Satellite | 54.71 | 66.06 | 4.03 | 67.48 | 56.97 | 33.62 | 72.65 | 2.07 | 1.56 | 67.22 | 68.59 | 6,434 | 36 |
| Pendigits | 8.11 | 28.34 | 53.81 | 25.82 | 18.26 | 2.66 | 14.78 | 57.40 | 57.11 | 13.26 | 14.56 | 6,870 | 16 |
| Annthyroid | 22.98 | 32.78 | 22.12 | 23.20 | 17.62 | 16.07 | 15.97 | 0.10 | 19.97 | 18.80 | 18.75 | 7,200 | 6 |
| MNISTv2 | 40.62 | 28.38 | 38.05 | 11.14 | 21.71 | 35.96 | 40.12 | 22.97 | 43.01 | 40.64 | 40.16 | 7,602 | 78 |
| Mammo. | 14.62 | 16.39 | 12.29 | 15.38 | 45.40 | 4.34 | 4.42 | 0.56 | 0.49 | 16.30 | 17.03 | 11,182 | 6 |
| Magic | 73.36 | 63.59 | 34.75 | 61.93 | 58.32 | 52.03 | 41.35 | 33.54 | 30.33 | 64.05 | 64.75 | 19,020 | 10 |
| Campaign | 27.03 | 26.52 | 75.52 | 35.27 | 37.16 | 11.21 | 32.54 | 21.84 | 70.81 | 26.88 | 27.54 | 41,188 | 62 |
| Shuttle | 17.90 | 94.62 | 35.49 | 96.57 | 96.06 | 9.95 | 50.95 | 8.72 | 16.34 | 91.62 | 91.30 | 49,096 | 9 |
| STMP | 28.33 | 0.47 | 14.83 | 0.46 | 0.42 | 0.46 | 51.51 | 6.85 | 20.65 | 35.82 | 38.99 | 95,156 | 3 |
| Backdoor | 41.13 | 3.76 | 66.30 | 5.15 | 6.86 | 48.94 | 39.33 | 1.52 | 1.66 | 50.50 | 50.39 | 95,328 | 193 |
| CelebA | 3.92 | 4.24 | 2.67 | 8.82 | 9.48 | 1.55 | 4.72 | 1.55 | 2.25 | 10.04 | 9.28 | 202,598 | 39 |
| Skin | 24.63 | 24.77 | 4.07 | 23.46 | 17.83 | 20.28 | 30.67 | 4.67 | 3.68 | 21.95 | 21.83 | 245,056 | 3 |
| Fraud | 9.28 | 13.19 | 4.71 | 24.22 | 23.65 | 8.53 | 20.80 | 1.83 | 10.90 | 17.19 | 16.84 | 284,806 | 29 |
| Cover | 3.81 | 4.15 | 7.24 | 2.47 | 6.53 | 0.62 | 8.63 | 5.39 | 7.05 | 13.53 | 9.21 | 286,048 | 10 |
| Census | 9.21 | 7.54 | 18.15 | 7.17 | 8.81 | 8.38 | 5.75 | 74.58 | 20.02 | 9.65 | 9.78 | 299,284 | 500 |
| Http | 1.17 | 97.44 | 2.38 | 35.80 | 30.13 | 3.67 | 33.25 | 2.36 | 2.68 | 47.70 | 47.04 | 567,498 | 3 |
| Donors | 12.04 | 11.56 | 50.98 | 11.87 | 20.87 | 6.11 | 5.92 | 10.88 | 100.00 | 32.35 | 31.91 | 619,326 | 10 |
| Median AP | 29.15 | 34.30 | 24.16 | 33.13 | 34.55 | 15.26 | 25.70 | 8.56 | 19.99 | 34.88 | 37.73 | | |
| Median Rank | 6 | 4.5 | 7.5 | 5 | 6 | 9 | 7 | 10 | 7.5 | **4** | **4** | | |

updates, where $M$ is the number of parameters in the network, and $N$ are the additional per sample parameters that serve as the anomaly score. Since modern NNs are typically overparametrized, we argue that the additional $N$ parameters do not limit the use of our method since typically $M > N$. Moreover, our approach can scale to extremely large datasets since training is typically performed in small batches. Across all examples used in this paper, a single run of our method does not take more than several minutes for the low-dimensional datasets and hours for the high-dimensional ones.

To further demonstrate the scalability of our approach, we provide, in Table 7, a run time comparison between PRAE and several baselines on the large datasets. In this experiment, we restrict our evaluation to CPU computing, thus providing a fair comparison to classical algorithms. As indicated by our results, the training of HBOS is highly effective and requires less training time than all baselines. On the other hand, our model scales better to larger and high-dimensional data in terms of inference time.

## F MNISTV1 AND FASHION MNIST

This section describes additional experiments performed using MNISTv1 and Fashion MNIST.

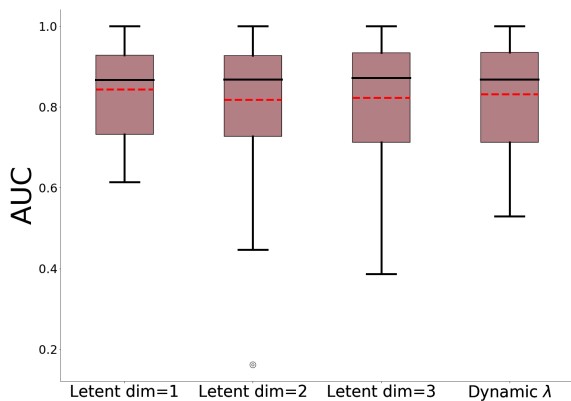

Figure 9: Box plots comparing the performance of PRAE for different latent dimension. In the right box plot we evaluate the dynamic procedure proposed for tuning $\lambda$ based on the MSE on unseen samples (see Section 5.2).

| dataset | Training [sec] | | | | | Inference [sec] | |
| --- | --- | --- | --- | --- | --- | --- | --- |
| | IForest | HBOS | NTL | GOAD | PRAE-$\ell_0$ | PRAE-$\ell_0$ | N/D/ Out % |
| Campaign | 2.3 | 1.9 | 712.7 | 9681.5 | 5.3 | 0.03 | 41188/ 62/ 11.3 |
| Shuttle | 1.1 | 1.9 | 673.1 | 5870.5 | 8.4 | 0.01 | 49097/ 9/ 7.1 |
| Backdoor | 18.2 | 2.4 | 1473.7 | 6225.8 | 6.4 | 0.11 | 95329/ 196/ 2.3 |
| CelebA | 8.8 | 2.7 | 3486.2 | 8741.2 | 25.4 | 0.18 | 202599/ 39/ 2.2 |
| Credit | 7.6 | 2.1 | 4105.9 | 6735.5 | 28.55 | 0.11 | 284807/ 16/ 0.2 |
| Census | 141.8 | 5.9 | 4048.2 | 20222.8 | 55.64 | 0.61 | 299285/ 500/ 6.2 |
| Donor | 17.2 | 2.5 | 9213.4 | 33734.3 | 34.36 | 0.19 | 619326/ 10/ 5.9 |

Table 7: Run time comparison between different baselines. The left part of thetable indicates training time, and our inference time is indicated in the right most column.

In this section, we provide an extended comparison with baselines. These include:

**Clustering-Based Local Outlier Factor (CBLOF) [He et al., 2003]** is a proximity-based method that relies on clustering to define an anomaly score for each sample.

**Angle-Based Outlier Detection (ABOD) [Kriegel et al., 2008]** uses vectors defined between pairs of points to identify outliers by comparing the angles between the different vectors.

**Connectivity-Based Outlier Factor(COF) [Tang et al., 2002a]** relies on proximity between samples to identify outliers.

**Subspace Outlier Detection (SOD) [Kriegel et al., 2009a]** identifies anomalies as samples that deviate significantly from the subspace spanned by a subspace defined based on neighbors of point.

**Locally Selective Combination of Parallel Outlier Ensembles (LSCP) [Zhao et al., 2019a]** is an ensemble method that uses multiple local subspaces to identify outliers.

**One-Class Support Vector Machines (OC-SVM) [Schölkopf et al., 2001]** uses support vectors to identify the margins of the normal part of the data. Here, we use a Gaussian kernel to capture the non-linearity of this method.

We omitted these methods from our results in the main text since the underperformed on the tabular datasets.

## F.1 SMALL MNIST DATASET (MNISTV1)

MNISTv1 was proposed in [Zhou and Paffenroth, 2017] for anomaly detection. To construct MNISTv1, we mix $4859$ nominal instances of the digit '4' with $265$ anomalies randomly sampled from all other digits. Following [Zhou and Paffenroth, 2017], we use a linear AE with one hidden layer of size $24$. Evaluation of the AUC of our method compared to all baselines appears in Table 3. This example appears to be especially challenging for density/distance-based baselines; we believe that this is due to the relatively high dimensionality of this data. In Figure 10, we present the $25$ most *inlaying* images (left panel), and the $25$ most *outlying* images (center panel) as identified PRAE-$\ell_0$. The identified inliers share a standard "simple" structure of the digit '4'. On the other hand, most identified outliers are indeed of different digits, except

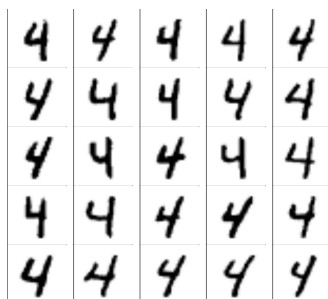 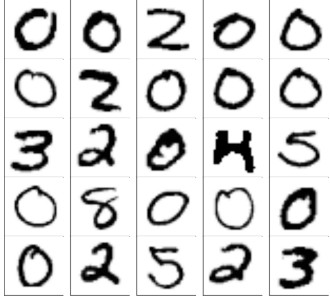 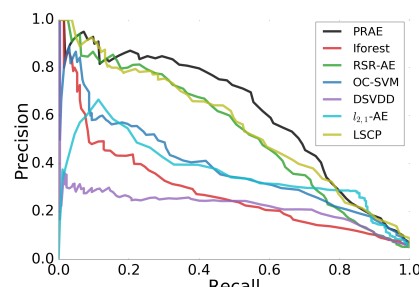

Figure 10: 25 most inlaying/outlying (left/center) MNISTv1 images as identified by PRAE. Right- precision vs. recall curves for leading baselines on the MNISTv1 data.

for one example, which is somewhat of an unusual instance of the digit '4'.

To further assess the performance of the different baselines on MNISTv1 we present the 25 most *inlaying* images and the 25 most *outlying* images as identified by the different baselines. These results are presented in Figures 11, 12, and 13.

Next, we repeat the MNISTv1 experiment using different classes as the representatives for the normal samples. Specifically, we mix 4859 nominal instances of the digit $C \in \{0, 1, ..., 9\}$ with 265 anomalies randomly sampled from all other digits. We compare the AUC of PRAE to two leading baselines across all classes. As indicated in Table 8, the proposed approach leads to more accurate outlier identification compared to the leading baselines across most of the classes.

| Class | Iforest | OC-SVM | PRAE-$\ell_0$ | PRAE-$\ell_1$ |
|---|---|---|---|---|
| 0 | 92.34 | 93.34 | 93.38 | 93.84 |
| 1 | 99.16 | 98.2 | 98.21 | 98.24 |
| 2 | 69.73 | 75.72 | 81.12 | 81.18 |
| 3 | 78.02 | 81.27 | 83.42 | 83.51 |
| 4 | 86.66 | 88.21 | 88.43 | 88.39 |
| 5 | 73.49 | 74.79 | 83.94 | 83.96 |
| 6 | 87.75 | 92.17 | 93.62 | 93.71 |
| 7 | 90.49 | 90.79 | 91.77 | 91.81 |
| 8 | 82.87 | 82.47 | 83.31 | 83.34 |
| 9 | 86.99 | 91.46 | 93.24 | 93.33 |
| T-shirt/top | 90.45 | 90.62 | 90.65 | 90.84 |
| Trouser | 97.77 | 97.83 | 97.96 | 97.90 |
| Pullover | 86.26 | 85.27 | 87.24 | 86.66 |
| Dress | 93.7 | 94.56 | 95.28 | 95.14 |
| Coat | 91.95 | 91.17 | 92.47 | 92.15 |
| Sandal | 92.12 | 91.58 | 91.14 | 91.15 |
| Shirt | 80.48 | 80.25 | 80.86 | 80.88 |
| Sneaker | 98.14 | 98.24 | 98.23 | 98.26 |
| Bag | 87.14 | 84.42 | 84.49 | 85.58 |
| Ankle boot | 97.65 | 98.16 | 97.95 | 97.91 |
| **Average** | 88.15 | 89.03 | **90.34** | **90.39** |

Table 8: Performance comparison with two leading baselines (IForest and OC-SVM) on the MNISTv1 and Fashion MNIST datasets. The top ten rows correspond to different classes in MNIST; each row indicates the class label used to define the normal samples. The bottom ten rows correspond to different classes in Fashion MNIST.

## F.2 FASHION MNIST

To evaluate the ability to identify outliers in Fashion MNIST, we mix 5000 nominal instances from the randomly selected 'Coat' class with 300 anomalies sampled from all other fashion items. Evaluation of the AUC of our method compared to

all baselines appears in Table 3. Next, we repeat this experiment using other classes as the majority/normal samples. In Table 8 we present the AUC of PRAE and two leading baselines across all classes. As evident from our result, the proposed approach outperforms leading baselines across most classes of Fashion MNIST.

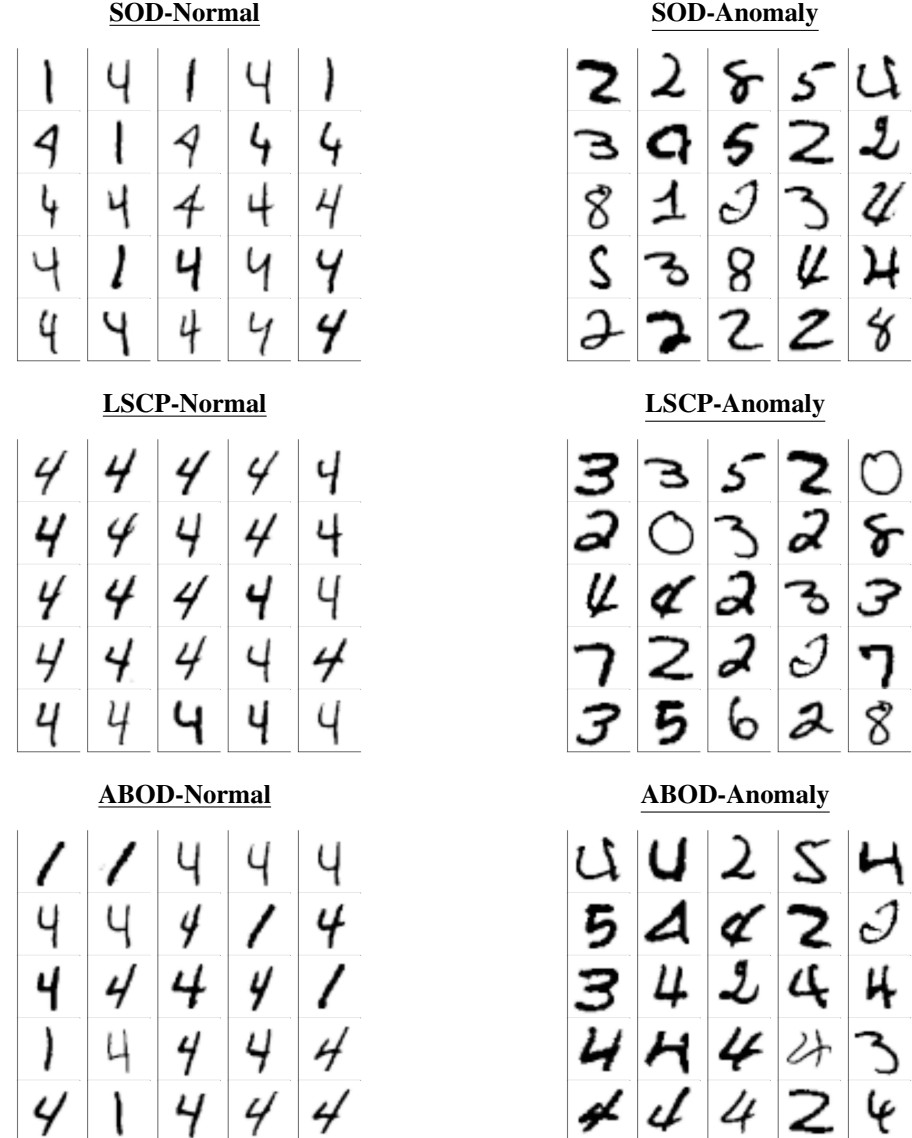

Figure 11: 25 most inlaying/outlying (left/right) MNISTv1 images as identified by different baseline algorithms.

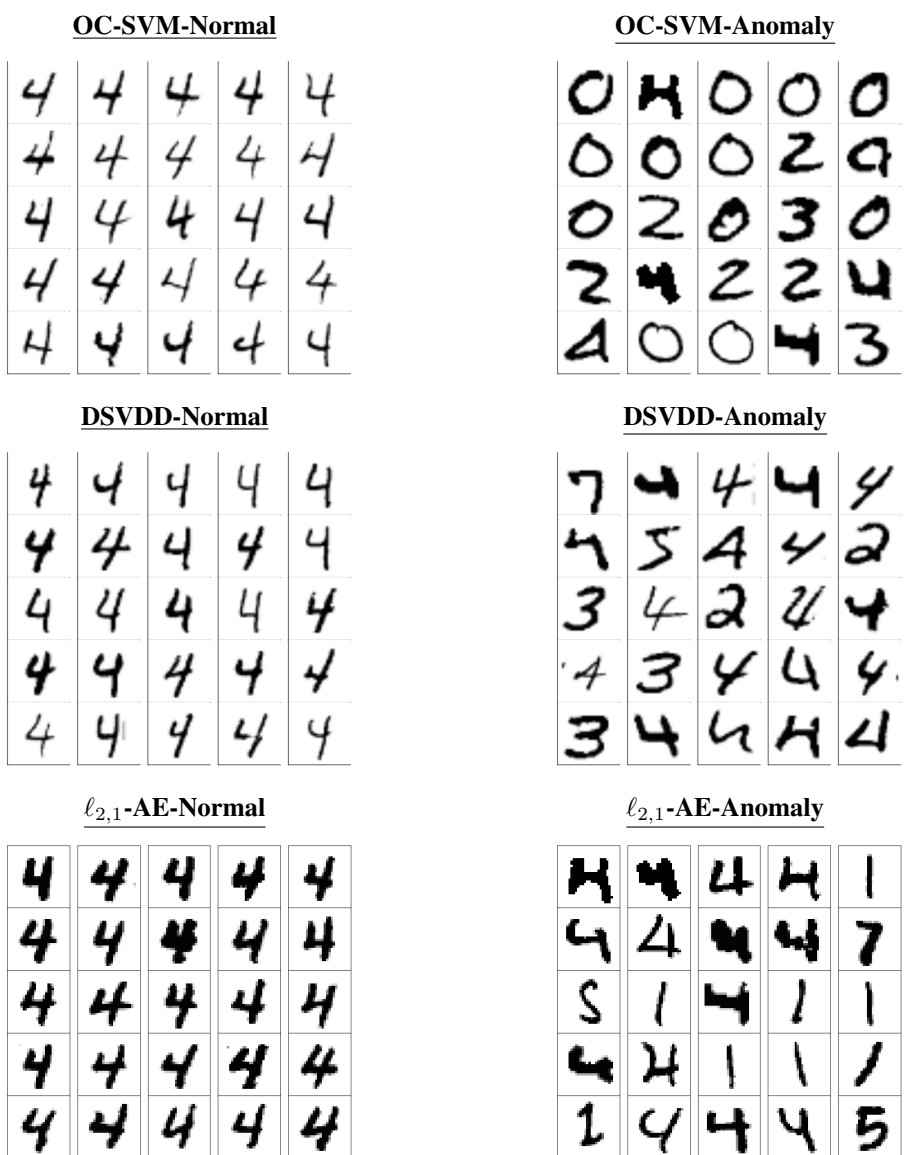

Figure 12: 25 most inlaying/outlying (left/right) MNISTv1 images as identified by different baselines

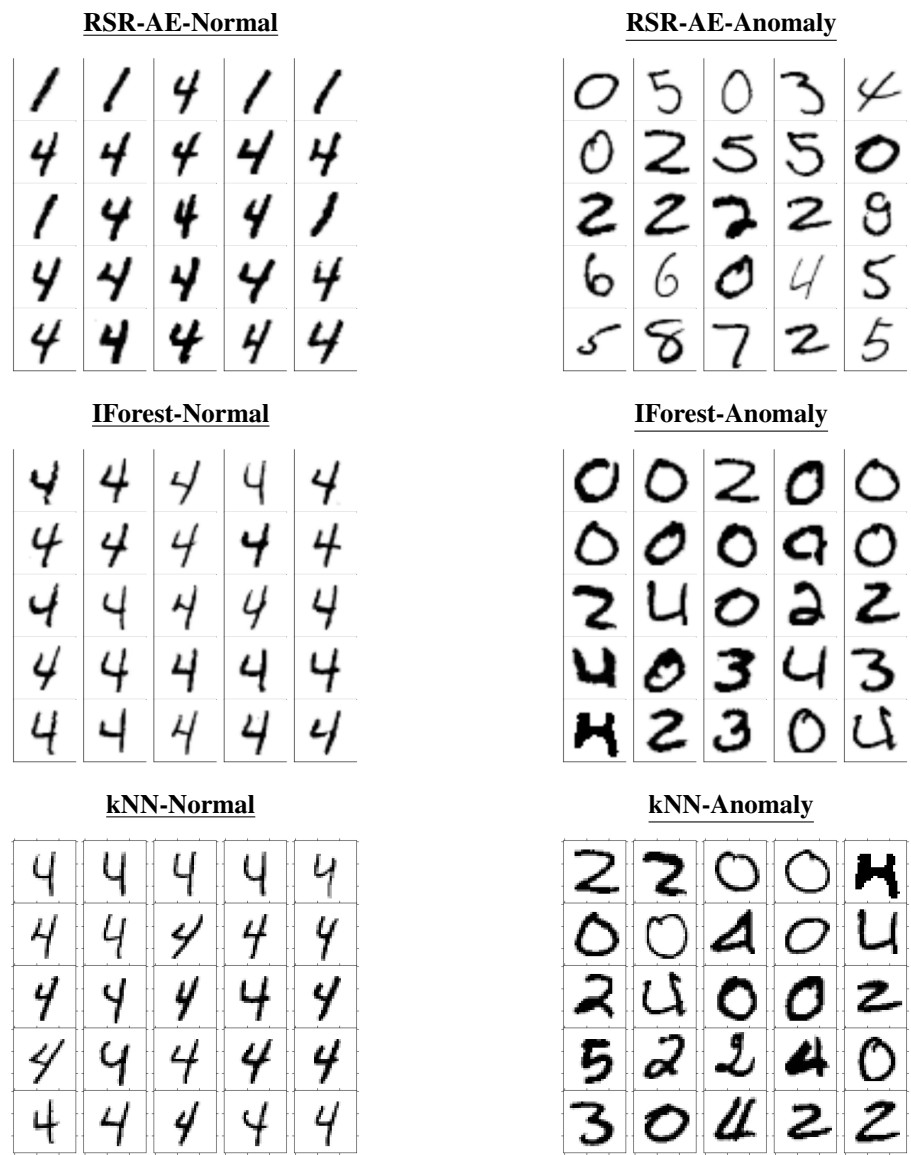

Figure 13: 25 most inlaying/outlying (left/right) MNISTv1 images as identified by different baselines