# OpenReview forum: "Transductive and Inductive Outlier Detection with Robust Autoencoders"
_auai.org/UAI/2024/Conference — UAI 2024 poster_

### Official Review · Reviewer_vUNN · 2024-03-11

**Q2-1 Originality-Novelty:** 3
**Q2-2 Correctness-Technical Quality:** 3
**Q2-5 Clarity Of Writing:** 3

**Q1 Summary And Contributions:**

In a "multivariate outlier detection scenario", by means of an AutoEncoder architecture, the authors propose a methodology to remove outliers during training, trying to complete the training with only inlier samples; after the model training, it is used for detecting outliers in new data (transductive + inductive).

**Q2-3 Extent To Which Claims Are Supported By Evidence:**

3: Good: the main claims are supported by convincing evidence (in the form of adequate experimental evaluation, proofs, (pseudo-)code, references, assumptions).

**Q2-4 Reproducibility:**

3: Good: key resources (e.g. proofs, code, data) are available and key details (e.g. proofs, experimental setup) are sufficiently well-described for competent researchers to confidently reproduce the main results.

**Q3 Main Strengths:**

A set of loss functions are proposed to do the training, in order to deal with the exposed issue (removing outliers in training).
The methodology avoids a combinatorial search over possible inliers by a probabilistic setting .

Theoretical guarantees of the behaviour of the proposed approach in Section 4.

Interesting experimentation on a synthetic dataset. Extensive experimentation on real datasets, and comparison with a other outlier detection approaches.

While it is true that a huge related literature can be found proposing algorithms to solve this problem, I consider that the paper has new and interesting ideas.

**Q4 Main Weakness:**

When working with AutoEnconders, they require that training data belongs to a single class-label, that it, a OneClass domain: I recommend the authors to remark this issue.

Read the last sentence in the second column, first paragraph, page 2: "inductive" and "transductive" concepts are used in the opposite way.

I miss a computational order analysis of the proposed pipeline.

The experimentation on the synthetic dataset and subsequent analysis interesting. I would recommend to enlarge synthetic experimentation over more datasets of this type, even reducing the number of experimented real datasets. In this setting, I think that results on synthetic datasets give more clues than real ones.

The structure of the AutoEncoder is fixed for any problem: is not natural to adjust this structure to, depending on the problem, its complexity, dimensionality, number of samples, etc.?

I consider that the statistical comparison between algorithms in the set of problems should be conducted by the "multiple comparison" setting proposed in
https://jmlr.org/papers/v7/demsar06a.html

**Q5 Detailed Comments To The Authors:**

Please check previous Q3 and Q4.

**Q9 Complying With Reviewing Instructions:**

Yes

---

> ### Author Rebuttal · Authors · 2024-04-07
>
> We thank the reviewer for the detailed and constructive comments, for appreciating our novelty, the strength of empirical results, the importance of the theoretical analysis, and for acknowledging that the interesting ideas provided here have value for the community. Below, we address all comments raised by the reviewer.
>
> One class domain: thanks for this point; we will clarify this in the main text.
> Inductive and transductive: thanks for spotting this; we will update our manuscript.
>
> Computational complexity: Please see Table 7 (in the Appendix), in which we show a detailed comparison between our model's runtime and different baselines in training and inference. We will add a summary of this result in the main text.
>
> Synthetic experiments: Thanks for this suggestion. We agree; in the appendix, we have included an additional synthetic experiment with a nonlinear manifold embedding in a high-dimensional space. If the paper is accepted, we will use the additional space allocated by UAI to move this experiment to the main text. Furthermore, following this suggestion, we will add an evaluation with synthetic data and different types of anomalies (local, global, cluster) [1].
>
> [1] Han, Songqiao, et al. "Adbench: Anomaly detection benchmark." Advances in Neural Information Processing Systems 35 (2022): 32142-32159.
>
> Multiple comparisons: This is a good suggestion; we will add these types of plots to compare our method to all baselines in the real-world setting.
>
>
> Again, we thank the reviewer for noting that our ideas are interesting. We are keen to improve our paper; therefore, we would be happy to respond if there are any additional open issues.

---

### Official Review · Reviewer_4Hnn · 2024-03-22

**Q2-1 Originality-Novelty:** 2
**Q2-2 Correctness-Technical Quality:** 3
**Q2-5 Clarity Of Writing:** 3

**Q1 Summary And Contributions:**

This paper proposes a method for outlier detection. The proposed method simultaneously learns the features and detects outliers based on autoencoder. Numerical experiments on a variety of datasets demonstrate the usefulness of the proposed method.

**Q2-3 Extent To Which Claims Are Supported By Evidence:**

3: Good: the main claims are supported by convincing evidence (in the form of adequate experimental evaluation, proofs, (pseudo-)code, references, assumptions).

**Q2-4 Reproducibility:**

3: Good: key resources (e.g. proofs, code, data) are available and key details (e.g. proofs, experimental setup) are sufficiently well-described for competent researchers to confidently reproduce the main results.

**Q3 Main Strengths:**

- Detecting outliers while learning the model

- Thorough experiments on 41 real datasets.

- Some theoretical analysis to a relaxed version of the loss function to which gradient methods are applicable.

**Q4 Main Weakness:**

- Unclearness for the optimization of the $\ell_0$ norm in Eq.(3)

- Influence of $\sigma$ is not investigated.

**Q5 Detailed Comments To The Authors:**

This paper proposes a method for outlier detection, which detects outliers in learning the model. The proposed method can be a simple extension of autoencoder, but I believe that this extension is effective. The first formulation of the loss in Eq.(1) is difficult to optimize. Therefore, a relaxed version of Eq.(1) is proposed, and theoretical analysis reveals that  the solution of the relaxed version is the same as Eq.(1). The proposed method is evaluated on a variety of real datasets.

If my understanding is correct, the $\ell_0$ norm in Eq.(3)  is optimized. I think that how to optimize it is not trivial, and some particular optimization method should be required. It is better to describe more details for optimization.

Another question is the influence of $\sigma$ in Eq.(2). This parameter should also affect the performance of the proposed method. The influence to the performance should be investigated.

**Q9 Complying With Reviewing Instructions:**

Yes

---

> ### Author Rebuttal · Authors · 2024-04-07
>
> We appreciate the reviewer's detailed and constructive comments, which acknowledge the novelty of our work and the strength of our empirical results. Below, we address all of the reviewer's comments.
>
> **L0 optimization-** We point out the reviewer to the left column of P3's last paragraph, in which we detail that: “We note that the regularization terms $E(∥z∥_0) =\Sigma P(z[i] > 0), and E(∥z∥_1) = \Sigma E(z[i])$ are parametric, and the expected value of the left term of Eqs. (3) and (4) is approximated using Monte Carlo sampling. Then, we differentiate the loss using SGD to update the weights in ψ and ρ, and the vector µ.”
>
> Specifically, we are dealing with probabilistic gates and computing the expectation of the $\ell_0$ norm by definition. This makes the regularization a smooth and fully differentiable function of $\mu$, which we can easily derive using gradient descent. The expectation  $E(∥z∥_0)$ becomes
>
> $\sum_{i=1}^N {P}(z[i] > 0) = \sum_{i=1}^N\left(\frac{1}{2} - \frac{1}{2} erf \left(-\frac{\mu_i}{\sqrt{2}\sigma}\right) \right)$, where $erf$ is the Gaussian error function.
>
>  A similar approach has been used in [1] and [2], and we will provide further clarification in the main text.
>
>
> Specifically, we are dealing with probabilistic gates and computing the expectation of the $\ell_0$ norm by definition. This makes the regularization a smooth and fully differentiable function of $\mu$, which we can easily derive using gradient descent. This approach has been used in [1], [2], and we will also clarify this in the main text.
>
> [1] Yamada, Yutaro, et al. "Feature selection using stochastic gates." International conference on machine learning. PMLR, 2020.
>
> [2] Louizos, Christos, Max Welling, and Diederik P. Kingma. "Learning sparse neural networks through $ L_0 $ regularization." arXiv preprint arXiv:1712.01312 (2017).
>
> **Choice of $\sigma$-** We have chosen our value of $\sigma$ based on prior research [1]. In our initial analysis, we found that our model performs well for values between 0.5 and 1.5. However, we acknowledge the reviewer's suggestion that further research on determining the optimal value for this parameter would be valuable for future work.
>
>
> **Simplicity and contribution-** we thank the reviewer again for appreciating our method's simplicity and its practical application value.
>
> **Optimization of $\ell_0$ norm-** please see our response above. After taking the expectation with respect to z, the $\ell_0$ norm is no longer an issue, and the loss becomes fully differentiable.
>
> **$\sigma$-** please see our response to this point above.
>
> We are keen to improve our paper and hope we have addressed all the reviewers' comments. We would happily respond within the remaining discussion time if there are still open issues.

---

### Official Review · Reviewer_iSnT · 2024-03-22

**Q2-1 Originality-Novelty:** 3
**Q2-2 Correctness-Technical Quality:** 3
**Q2-5 Clarity Of Writing:** 3

**Q1 Summary And Contributions:**

This paper introduces a robust method employing an autoencoder to detect outliers both during training and testing phases. Additionally, it proposes an approximation of the architecture and empirically validates the tightness of this approximation through experiments. This approximation significantly enhances training efficiency. Furthermore, the paper presents an unsupervised scheme for tuning hyperparameters.

**Q2-3 Extent To Which Claims Are Supported By Evidence:**

2: Fair: the main claims are somewhat supported by evidence (but the experimental evaluation may be weak, or does not match entirely with the claims, important baselines may be missing, proofs contain important ideas but lack rigor, algorithmic details are only discussed superficially, references are imprecise, assumptions are not sufficiently motivated or explicated, etc.).

**Q2-4 Reproducibility:**

3: Good: key resources (e.g. proofs, code, data) are available and key details (e.g. proofs, experimental setup) are sufficiently well-described for competent researchers to confidently reproduce the main results.

**Q3 Main Strengths:**

The paper presents a valid theoretical framework supported by empirical evidence on synthetic datasets. Additionally, it offers detailed experiments conducted across various data structures, providing comprehensive insights into the proposed methodology.

**Q4 Main Weakness:**

The robustness of the approach hinges on the selection of the hyper-parameter lambda, which has proven to be insensitive in real data experiments. However, it's important to note that this insensitivity doesn't extend to lambda values greater than 1. In the synthetic data experiments, the author observed that lambda values exceeding 1 often lead to undesirable outcomes. To verify whether this phenomenon does not exist in real data experiments, further investigation is warranted.

Moreover, in scenarios where lambda exceeds 1 and yet yields favorable results, there arises a question regarding the efficacy of the loss function design. It's plausible that lambda values larger than one or close to one might contain all data points, rendering the loss function ineffective. On the other hand, when lambda equals 1, it could be useful to plot the number of data points included during training.

**Q5 Detailed Comments To The Authors:**

I suggest the author to include a discussion on the manifold assumptions underlying each dataset. This discussion could provide insights into the underlying structure of the data and how it influences model performance. Furthermore, I recommend marking out the best performance achieved for each dataset in a table. This will help readers quickly identify the most effective methods across different experiments, facilitating comparisons and understanding of the results.

**Q9 Complying With Reviewing Instructions:**

Yes

---

> ### Author Rebuttal · Authors · 2024-04-07
>
> We thank the reviewer for the detailed and constructive comments and for appreciating our novelty, technical, and writing qualities.
>
> Sensitivity/insensitivity to $\lambda$: This particular parameter plays a crucial role in determining how our model handles anomalies during training. We intentionally designed our model to be influenced by this parameter, meaning that when we use large values of $\lambda$, the model's performance as a robust autoencoder decreases - but this is actually desirable. If our model didn't behave this way, it would essentially become a standard autoencoder, rendering the impact of our induced regularizer insignificant.
>
>
> In Figure 3, we have used synthetic data to demonstrate that our model shows an expected "phase" transition when we change the value of $\lambda$ between acting as a robust AE and a standard AE. This transition is evident in both the F1 score and the MSE. Therefore, we are not sure why the reviewer is suggesting that "further investigation is warranted" to verify if this phenomenon exists in real data experiments. We actually want this phenomenon to exist, and we know what happens when $\lambda$ is much larger than 1. In this case, all the samples will be included, and our model will act as a standard AE. Notice that the phase transition is around $\lambda= 1$ in Figure 3 due to the fact that the data is normalized.
>
>
>
> We will add lambda values higher than 1 (but not too large) to Figure 5 to further demonstrate the stability of our model. We expect that our model will remain relatively stable for values close to one from above as well. In contrast, if we use values where $\lambda$ is much greater than 1, all samples will be included, and this will result in lower AUC values (as shown in Table 3).
>
> In the table below, we provide statistics regarding the portion of selected points with $\lambda=1$.
>
>
> Discussion on manifold assumption: We agree with the reviewer that it would be interesting to study the datasets where the manifold assumption holds a priori. This could help users understand which datasets our method would be most useful for. To address this, we used [1] to estimate the intrinsic dimension of all the datasets evaluated in the paper. Below is a subset of the results we obtained.
>
>
>
> | Datasets                | Vowels | Lymphography | Stamps | Musk | pendigits | donors | WBC  | PageBlocks | census | smtp | satellite | Letter |
> |-------------------------|--------|--------------|--------|------|-----------|--------|------|------------|--------|------|-----------|--------|
> | Portion of samples kept | 0.84   | 0.61         | 0.84   | 0.93 | 0.91      | 0.81   | 0.81 | 0.92       | 0.70   | 0.92 | 0.85      | 0.82   |
> | Intrinsic Dim           | 5.88   | 6.57         | 4.71   | 7.29 | 5.63      | 1.08   | 5.55 | 3.74       | 3.76   | 1.77 | 8.42      | 6.75   |
>
> Our results in the table above demonstrate that: (1) our model with $\lambda=1$ keeps most samples but not all. (2) the intrinsic dimension in most datasets is low, supporting the use of the proposed PRAE to learn the latent manifold and recover the outliers.
>
>
> Absence of discussion on the voice of $\lambda$: We deeply appreciate the reviewer's insightful comment. In our response, we aim to highlight the various sections where we elaborate on our selection of the single hyperparameter.
>
> P2 Left Column: We propose two unsupervised schemes to tune the regularization parameter, controlling the robustness of the PRAE.
>
>  P6, the entire subsection 5.2 dives into two novel unsupervised schemes for tuning this parameter, including several experiments.
>
> P7 left column “We use a single value of λ equal to 1 in all our experiments, based on the results of Section 5.2. We also evaluated the dynamic tuning of λ proposed in Section 5.2, and its results are presented in Figure 5.2. “
>
> P8 Figure 5 presents a sensitivity analysis demonstrating that the model works well for many values of $\lambda$ that are close to 1. And a positive result demonstrating that $\lambda=0$ hurts our model.
>
> Section D in the appendix provides additional information about all the hyperparameters.
>
> Given this extensive discussion about the choice of parameter $\lambda$, we are having a hard time understanding this comment by the reviewer.
>
>
>
>
> Finally, we believe that our paper includes many valuable independent contributions to the community. These include (1) a new method for robust autoencoders that is useful for robust subspace recovery and anomaly detection. (2) two unsupervised schemes for tuning the regularization parameter. (3) theoretical analysis comparing probabilistic robust AE to the deterministic robust AE. (4) extensive experimental results, including an ablation, synthetic linear, nonlinear, and 42 real-world datasets.
>
> We are keen to improve our paper and hope we have addressed all the reviewers' comments. We would happily respond within the remaining discussion time if there are still open issues.

---

### Meta-Review · Area_Chair_Gf23 · 2024-04-16

Reviewers acknowledge relevance of the problem and find the solution mostly convincing. They also appreciated theoretical guarantees as well as extensive evaluation with real data.